

# Observed currents in the Archipelago Sea

Hedi Kanarik[1], Laura Tuomi[1], Pekka Alenius[1], Elina Miettunen[2], Milla Johansson[1], Tuomo Roine[1], Antti Westerlund[1], and Kimmo K. Kahma[1]

[1] Finnish Meteorological Institute, P.O.Box 503 , FI-00101 Helsinki, Finland
[2] Finnish Environment Institute, Latokartanonkaari 11, FI-00790 Helsinki, Finland

**Correspondence:** Hedi Kanarik (hedi.kanarik@fmi.fi)

**Abstract.**

The Archipelago Sea (AS) in the Baltic Sea is a complicated fragmented sea area with numerous small islands and islets that are crossed by several deeper straits. The area functions as a major route for both transport and leisure activities on the sea as well as many other forms of blue economy. Even with the high maritime activities in this sea area, knowledge of the currents along the deep channels crossing the AS has been limited due to the lack of quality ensured measurements. To enhance the general understanding of the dynamics in the AS, we have collected and analysed currents in 10 different locations across the AS using ADCP (Acoustic Doppler Current Profiler) current measurements made during numerous short measurement campaigns over the last two decades. Currents in the AS are restricted by the geometry of the area and typically have two main flow directions, with a slightly wider directional distribution at the southern parts of the AS. In open areas of the AS, surface currents have a mean magnitude of around $8 \ \mathrm{cms}^{-1}$. Currents in the AS can momentarily grow to significant magnitudes, as even the shortest measurement time series (of around 4 months) have measured currents exceeding at least $40 \ \mathrm{cms}^{-1}$. The current magnitudes in the long and narrow straits are almost double ($14 \ \mathrm{cms}^{-1}$) the general current magnitudes of the area and current magnitudes up to $115 \ \mathrm{cms}^{-1}$ have been measured at the northern end of the long NE strait. The AS currents are mostly driven by local winds, but strong oscillations in the surrounding basing can cause significant current (up to $80 \ \mathrm{cms}^{-1}$ in narrow straits) if the local winds are weak enough (below $10 \ \mathrm{ms}^{-1}$) or from non optimal direction related to the straits. Our analysis shows that especially strong seiche in the Gulf of Finland combined with low sea levels in the Gulf of Bothnia is significant factor in forcing currents in the AS.

## 1 Introduction

The Archipelago Sea is a unique and complicated coastal archipelago located in the Baltic Sea (Fig. 1). It is a region with over 40 000, small islands and islets, narrow channels between islands, and a large variation in bottom topography. Together with the Åland Sea, the Archipelago Sea (AS) separates the two major Baltic Sea basins with independent oscillation and circulation patterns: the Baltic proper together with the Gulf of Finland from the Gulf of Bothnia (Wubber and Krauss, 1979; Jönsson et al., 2008). The AS is also a heavily trafficked area with narrow fairways. In some of these fairways there can be sudden strong currents (Kanarik et al., 2018), potentially causing dangerous situations for navigation, when the flow direction is from the side of the vessel. For safety of the marine traffic it is essential that different operators, e.g. pilots, have detailed



knowledge of areas and situations in which such strong currents can occur. Thus, understanding the main features of currents in different areas of the archipelago and the drivers behind them is important. As currents also transport substances within the water column, understanding circulation dynamics provides knowledge-based information to support spatial planning for offshore activities such as aquaculture.

The first current measurements in the Baltic Sea were conducted more than a century ago using a large network of light ships (Witting, 1912; Palmén, 1930; Hela, 1952). Studies based on these measurements gave us information on the general circulation patterns of the Baltic Sea and showed that the currents, especially in the surface layer, are strongly dependent on the wind climate. The general Baltic Sea wind fields are characterised by dominant south-westerly (SW) and secondary north-northwesterly (NNW) winds (Soomere and Keevallik, 2001; Männikus et al., 2020), as shown also in the AS area (Miettunen

et al., 2024). The Baltic Sea is located at the end of the North Atlantic storm track and has large seasonal variability in occurrence of windstorms, with most frequent occurrences between November and January, and least between May and August (Laurila et al., 2021a), while also experiencing large annual and decadal variability (Laurila et al., 2021b). These variations in wind and atmospheric pressure are also the main factors affecting short-term sea level variations in the Baltic Sea (Johansson, 2014). Recent studies have also been able to further identify that most extreme sea level events in the Baltic Sea are related to

events with clusters of cyclones crossing the Baltic Sea region in a short period of time (Rantanen et al., 2024).

    Today, only observation-based studies of the circulation in the Baltic Sea are rather rare and have been mainly focused on certain areas as part of temporary moorings utilising Acoustic Doppler Current Profiles (ADCPs) (e.g., Muchowski et al., 2023; Mattias Green et al., 2006). Several studies have recently focused on the Gulf of Finland, where analysis of measured currents has shown that while currents are mainly driven by wind and bathymetry, seiche and tides can play a significant role in the

area's circulation dynamics (Lilover et al., 2011). In the Gulf of Finland, current oscillations have been shown to be strongly aligned with seiche modes in different parts of the water column depending on the prevailing stratification (Suhhova et al., 2018). Seiche oscillation is a specific feature of the semi-enclosed Baltic Sea sea level variations. It typically forms when wind and air pressure induce strong sea level gradients between different sub-basins (e.g., Gulf of Finland – Baltic proper). After the atmospheric forcing ceases, strong gradient between basins can result in an oscillation of water back and forth between the

sub-basins. The periods of these oscillations range up to 27 hours between the Baltic proper and the Gulf of Finland, and up to 39 hours between the Baltic proper and the Gulf of Bothnia (Lisitzin, 1974; Wubber and Krauss, 1979). The tides in the Baltic Sea are usually of the order of a few centimetres (e.g., Witting, 1908, 1911). The strongest tidal oscillations have been observed in the Gulf of Finland with amplitudes of 17–19 cm (Medvedev et al., 2013), and are clearly distinguishable in the current spectrum (Lilover, 2012).

In the AS, the first current measurements were made in 1922 as part of a study related to water exchange studies between the Baltic proper and the Bothnian Sea (Witting, 1925; Lisitzin, 1951). Measurements made near the surface and at 20 m depth showed a strong steering of the currents due to the bathymetry and geometry of the islands. In 1977, current measurements were made at three locations in the Archipelago Sea (Ambjörn and Gidhagen, 1979). Those records included a strong northward current of 91 $\mathrm{cms}^{-1}$ at one of the measurement locations at the narrow south-northward strait, while the maximums at the two

other locations were around 50 $\mathrm{cms}^{-1}$. During the years 1974–1977, a large measurement campaign was carried out in the





inner part (closer to the mainland) of the AS (Virtaustutkimuksen neuvottelukunta, 1979) with the aim of evaluating the fate of substances released from wastewater sources. Measurements were made at 15 locations and showed that currents in the inner and mid archipelago have short-term fluctuations between two main directions. The current magnitude was weak on average, but occasionally there were high magnitudes. The highest measured current magnitude was c. 50 $\mathrm{cms}^{-1}$. In 2004, 2006 and from 2016 onwards there have been newer current measurement campaigns in the outer regions of the Archipelago Sea using Acoustic Doppler Current Profiles (ADCPs); however, they have been largely unpresented and published before this paper.

Earlier research on AS currents has focused mainly on the effect of the wind and the geometry of the area to explain the drivers of the current fields. Virtaustutkimuksen neuvottelukunta (1979) evaluated the effect of winds on currents in the inner parts of the AS and concluded that the interactions are extremely complicated due to the heterogeneity of the area. The connection between wind direction and current direction was clearest when the winds blow from the same direction for longer periods and are aligned along the channels. A model study, focussing on the more open areas of the AS, has shown that due to the geometry of the archipelago and the orientation of the channels, the southward transports were the largest with NNW winds and the northward transports with SSE–SE winds (Miettunen et al., 2020). The prevailing wind direction in the area, SW, is not optimal for northward or southward flow through the archipelago. This was also demonstrated by the analysis of current measurements in the Lövskär cross section, where currents were shown to be very sensitive to even small changes in wind directions and that even 15 $\mathrm{ms}^{-1}$ SW winds were not able to induce strong currents in this area (Kanarik et al., 2018). Recent modelling studies have shown that there is large spatial and year-to-year variability in current speeds and directions in the Archipelago Sea (Tuomi et al., 2018; Miettunen et al., 2020, 2024). The model results show that in the deep, narrow channels that cross the area in the N–S or NW–SE direction, the current speeds are highest and the directions are strongly aligned along the axis of the channels. In more open areas, the directional distribution of currents is wider and the currents are weaker.

In this paper, we publish ADCP datasets from the Archipelago Sea area and describe the general features of the measured currents. In our analysis, we considered the heterogeneity in the temporal and spatial coverage of the data set. Using some of the longer datasets, we examined the drivers of the strong currents in the AS, which were left unexplained in the earlier research that focused mainly on the local wind conditions (Virtaustutkimuksen neuvottelukunta, 1979; Kanarik et al., 2018). This was done by also considering the differences in sea level and atmospheric pressure over the AS.

## 2 Materials and methods

### 2.1 Current measurements

In this paper, we present and analyse only current measurements that are done with ADCP current profilers. We had data from 10 different locations around the Archipelago Sea (Table 1, Figure 1). Measurements were done using Teledyne RD Instruments' bottom-mounted 300 kHz WORKHORSE Sentinel Broadband Acoustic Doppler Current Profiler. Data were quality checked using internal quality parameters (signal correlation, echo intensity, percent good, and error velocity) based on recommendations by Book et al. (2007) and Symonds (2006). For more detailed information on the exact procedure and the





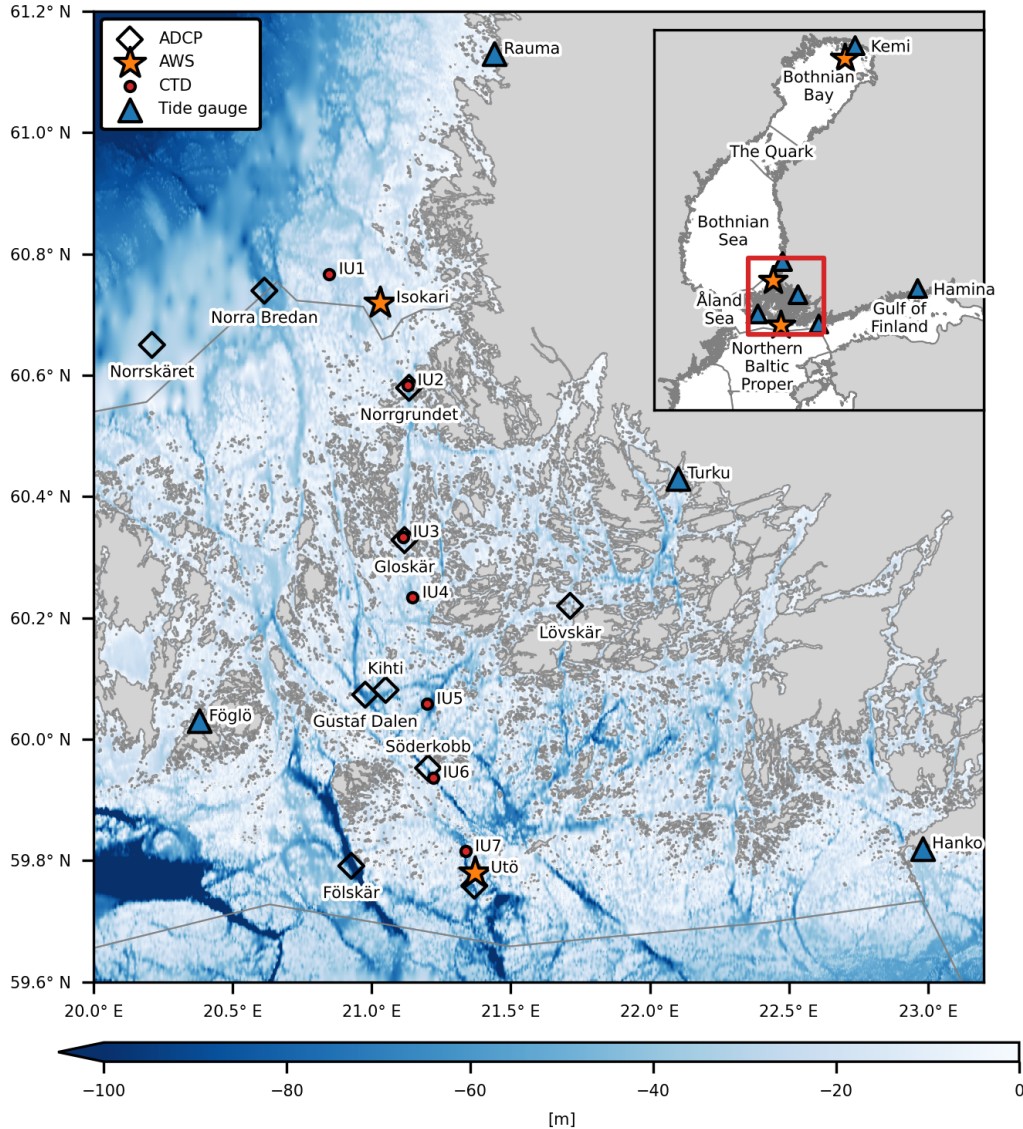

**Figure 1.** Locations of the current measurements (ADCP, acoustic Doppler current profiler) in the Archipelago Sea, indicated with diamonds. The stars and triangles indicate the locations of the automatic weather stations (AWS) and tide gauges, respectively. Red dots indicate the location of the routine CTD (conductivity–temperature–depth) stations between Isokari and Utö. The background colour represents the bathymetry (EMODnet Bathymetry Consortium, 2020). The smaller map shows the different basins in the northern Baltic Sea, and the red rectangle indicates the location of the study area. Basin borders and coastline from HELCOM (2018).

threshold values used, see Kanarik et al. (2018, Section 3). In the following analysis, we used only ensembles quality flagged
as 1 (good data).



**Table 1.** ADCP moorings used in this study. The given depth is the depth measured during the deployment of the ADCP.

| Name | ID | Measurement period | Location | Depth |
|------|-----|-------------------|----------|-------|
| Fölskär | SVT04 | 11 May – 11 Oct 2004 | 59°47.466' N, 20°55.596' E | 115 m |
| | SVM2S | 25 Apr – 25 Aug 2006 | | |
| Söderkobb | SVM1S | 25 Apr – 25 Aug 2006 | 59°57.15' N, 21°12.17' E | 60 m |
| Norrskäret | SVM2N | 27 Apr – 22 Aug 2006 | 60°39.000' N, 20°12.600' E | 49 m |
| Lösvkär | lov13 | 18 June – 13 Nov 2013 | 60°13.183' N, 21°42.800' E | 44 m |
| Norra Bredan | S1-16 | 6 Sept 2016 – 22 Feb 2017 | 60°44.358' N, 20°36.852' E | 68 m |
| Norrgrundet | S2-16 | 6 Sept 2016 – 16 Oct 2018 | 60°34.752' N, 21°08.046' E | 54 m |
| | S2-17 | device changed on 7 May 2017 | | |
| | S2-21 | 20 May 2021 – 3 Dec 2021 | | |
| Utö | uto17 | 26 July 2017 – 27 June 2018 | 59°45.510' N, 21°22.146' E | 76 m |
| Kihti | kih19 | 19 Nov 2019 – 23 May 2020 | 60°04.872' N, 21°03.000' E | 41 m |
| Gustaf Dalen | mame2020 | 14 Oct 2020 – 6 Sept 2022 | 60°04.399' N, 20°58.600' E | 66 m |
| | mame2021 | device changed on 10 Aug 2021 | | |
| Gloskär | IU3 | 20 May 2021 – 2 June 2022 | 60°19.690' N, 21°07.063' E | 50 m |

The amount of subsurface bins that were rejected due to side slope contamination varied between the measurement stations and are presented in Table 2 together with information about the depth of the first measured bin, bin size, ensemble interval, number of pings per ensemble and the corresponding standard deviation (std) of the measurements. As the sea surface variations in the Archipelago Sea (AS) are small, typically less than the bin size of the measurements, one depth level was used for

flagging the data for side slope contamination for each time series (cf. Last bin distance from the surface Table 2). ADCP was set to reach the sea surface in all but one data set: the Fölskär measurements from 2004, in which the topmost bin was at 22 m depth. In the used configurations, we lost data from the about five metre boundary layers at the sea surface and at the sea floor. Magnetic variation was taken into account during the deployment setup.

### 2.1.1 Quality issues

At some measurement sites and periods, there was loss of data due to physical or biogeochemical conditions (e.g., ice cover or lack of scatters). These are marked as "quality issues" in Table 2. The amount of good data for the whole water column for each station is presented monthly in Figure 2, where the values indicate the fraction of good data for the whole month from all of the measured depths. If measurements started or ended in the middle of the month, then only the fraction of the measured month is marked as good. For example, measurements at Kihti station started on 19 November and there were no major issues

in the measurements so the measurements covered about 40% of that month.

The ice cover affected the measurement quality in February and March 2019, 2021, and 2022 (Fig. 2). The effects of ice were notable on Feb–Mar 2018 in Norrgrundet, on March 2018 in Utö, and in Gustaf Dalen on two winters: Feb–Mar 2021 and



**Table 2.** Quality information about the ADCP measurements. *Check Sect. 2.1.1 for more precise description of the quality issues.

| Name | Bin size (m) | Ensemble interval (s) | Pings/Ens | std (cm/s) | 1st bin depth from the sea floor (m) | Last bin distance from the surface (m) | Quality issues* |
|---|---|---|---|---|---|---|---|
| Fölskär | 2 m | 1800 | 100 | 0.70 | 4.0 | 22 | depth and scatterers |
| – | 2 m | 1800 | 100 | 0.70 | 6.0 | 7 | scatterers |
| Söderkobb | 2 m | 1800 | 100 | 0.70 | 5.5 | 4 | - |
| Norrskäret | 2 m | 1800 | 100 | 0.70 | 5.0 | 4 | - |
| Lövskär | 1 m | 1200 | 120 | 1.24 | 4.7 | 5 | - |
| Norra Bredan | 1 m | 1800 | 60 | 1.75 | 4.8 | 4 | - |
| Norrgrundet | 1 m | 1800 | 65 | 1.68 | 4.8 | 5 | - |
| – | 1 m | 1800 | 65 | 1.68 | 8.8 | 5 | ice and bottom |
| – | 1 m | 1800 | 130 | 1.19 | 4.7 | 5 | - |
| Utö | 1 m | 1800 | 160 | 1.07 | 4.7 | 7 | ice and scatterers |
| Kihti | 1 m | 1200 | 120 | 1.24 | 4.8 | 4 | - |
| Gustaf Dalen | 1 m | 1200 | 140 | 1.15 | 4.9 | 5 | ice and scatterers |
| – | 1 m | 1200 | 140 | 1.15 | 4.6 | 5 | ice and scatterers |
| Gloskär | 1 m | 1800 | 229 | 0.91 | 4.6 | 4 | - |

Dec 2021–Mar 2022 (Fig. 2). At the Utö and Gustaf Dalen sites, ice affected the measurements mainly only in the upper half of the water layer. However, in Norrgrundet, the ice corrupted the measurements in the entire water column and the data were discarded for the whole time period from February 6 to March 22, 2018. During March 2018, also most of the measurements at the Utö station were bad quality.

The lack of scatterers in spring and early summer also caused reoccurring quality issues in the ADCP data. Because of this, the ADCP did not receive enough valid signals from certain depths of the water column, typically around pycnocline. At Gustaf Dalen, ADCP was unable to measure currents around 10–2 m depths during May 2021 (Fig. 2). Data were also missing at the Utö station in the beginning of June 2018 at depths of 10–24 m and in the topmost 10 m layer in the last half of June. In some areas, measurements were successful during the night, when zooplankton migrates to the surface. An example of this lack of data in parts of the water column can be seen in Westerlund et al. (2022, Fig. 4). Zooplankton movement also caused some minor issues, e.g. in the thermocline in the Lövskär dataset (Kanarik et al., 2018), but only a nominal part of the data had to be rejected.

The depth of the Fölskär site (115 m) was too deep for the 300 kHz ADCP as the devise was mostly unable to acquire measurements in the upper 50–60 m of the water column. In the first data set (measured in 2004), the maximum number of measurements was missed at depths of 36–38 m, where ADCP was unable to compute the velocity solution from 68% of the ensembles. The next measurement campaign in 2006, was only slightly more successful with the maximum amount of missing



data being 53% at 36–38 m depth. The lack of scatterers also contributed to the amount of missing data in this site. Data

were missing most frequently between June and July in both years. Measurements close to the surface (7 m depth, Table 2) were only obtained in 2006 from mid-June onwards. Together, of the two measurement campaigns in Fölskär, only 35% of the measurements were successful at the near-surface layer above the seasonal thermocline.

In the second Norrgrundet dataset, vertical velocities and thus error velocities were exceptionally high at the bottommost 4 bins. The error velocity tests rejected 21% of the data from these depths, but since the vertical velocities still had unrealistic

data after these automatic quality checks, we rejected all measurements from the four bottommost depth bins in this data set.

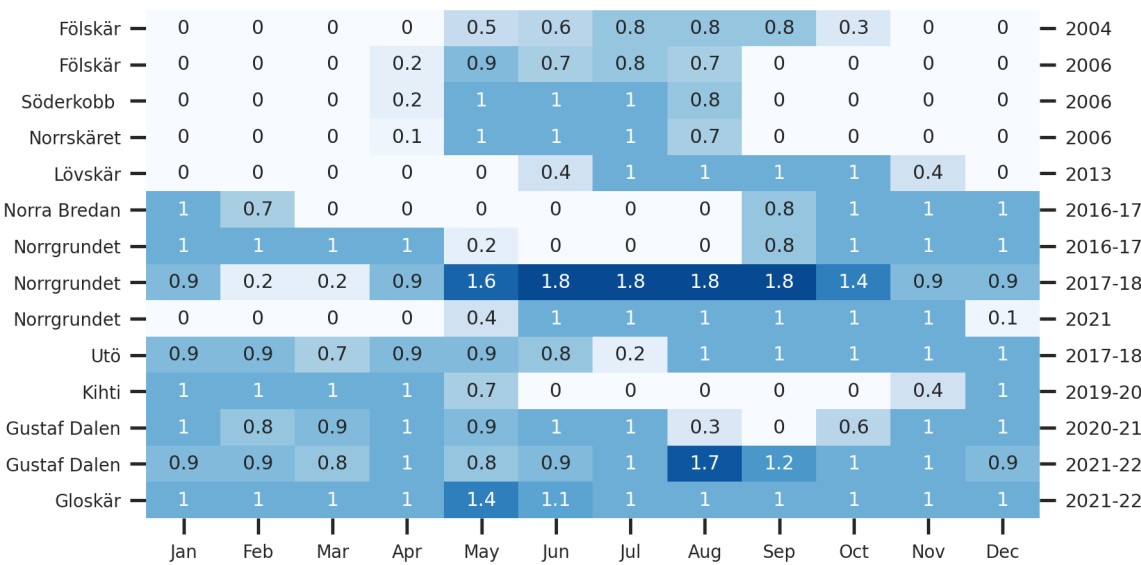

**Figure 2.** Amount of good data available throughout the whole water column per month for each measurement dataset (Table 1). The years of the measurements are shown on the right-hand side y-axis. Value 1 indicates that data were good throughout the water column for a whole month and currents were measured at this location only during one year. Values greater than 1 indicate that the measurements spanned the next year. See Table 1 for the exact measurement periods.

## 2.2 Wind, sea level and hydrography data

We used wind data from two automated weather stations (AWS) to represent weather conditions in the surrounding seas. Utö AWS (59°46' 44.74" N 21°22' 29.24" E) is located on the southernmost island of the AS and Isokari AWS (60°43' 19.91" N, 21°1' 36.52") at the northeast edge of the AS (locations shown in Fig. 1). Easterly, southerly and westerly winds (measured

at 25 m above mean sea level) at the Utö AWS represent the conditions in the northern Baltic proper, but northerly winds are affected by the northern islands and NW peninsulas of Utö island. Isokari AWS has open sea from SW to N and the winds are measured at 31 m above mean sea level. The size and shape of the Isokari island weaken the winds from NNE to S directions; however, winds from E are slightly overestimated due to the height of the measurement device. For analysis of larger-scale



atmospheric conditions, atmospheric pressure measurements from Kemi lighthouse AWS (65°23' 6.28" N, 24°5' 44.46", Fig.
1) were used together with Uto AWS measurements.

For statistical analysis of general wind conditions over AS, we used climatological standard normal period of 1991 to 2020.
For the first decade, data was available only once every 3 hours. Thus, statistical analysis over given 30 year period was done
using the same time interval of 3 hours. The highest available temporal resolution was used when the winds were analysed
against simultaneous current measurements. Measurements with 10-minute intervals were available in Utö AWS starting from
September 13, 2006 and in Isokari AWS from May 2, 2006. Continuos simultaneous data of atmospheric pressure from multiple
ASW stations were available only from 2007 onwards, and the statistical analysis on this variable was thus conducted over a
shorter time period.

To evaluate the relation between currents and the sea level differences over the area, we analysed hourly instantaneous sea
level data from tide gauges at Rauma (61°8' 2.03" N, 21°26' 33.47" E), Hanko (59°49' 22.32" N, 22°58' 35.70" E), Turku
(60°25' 41.81" N, 22°6' 1.92" E), Föglö Degerby (60°1' 54.78" N, 20°23' 5.34" E), Hamina (60°33' 45.96" N, 27°10' 45.12"
E) and Kemi (65°40' 24.13" N, 24°30' 54.94" E). The locations are marked in Figure 1 with triangles. As in this study, we
are interested in the sea level tilt driving the currents, we used hourly sea levels relative to the theoretical mean sea level for
statistical analysis over climatological standard normal period of 1991 to 2020 and half-hourly measurements for analysis on
simultaneous current and sea level measurements.

Wind and sea level measurements are openly available in the FMI's open database (https://en.ilmatieteenlaitos.fi/open-data).

Temperature and salinity profiles were collected from 7 stations located along a north–south transect across the Archipelago
Sea from Isokari to Utö (Fig. 1, IU locations). Data were analysed for years 1991 to 2021 during which the total number of
profiles measured at each station varied between 35 and 66. This data is available through https://www.marinefinland.fi/en-US
or through the ICES data system (https://www.ices.dk).

**2.3   Calculation of persistency (P)**

The variability of the current direction was described using the persistency value (P) as defined by Witting (1912) and Palmén
(1930):

$$P = \frac{\sqrt{(\frac{1}{N}\sum_n u_n)^2 + (\frac{1}{N}\sum_n v_n)^2}}{\frac{1}{N}\sum_n \sqrt{u_n^2 + v_n^2}} \times 100, \tag{1}$$

where $u$ and $v$ are the eastward and northward velocity components of the currents, respectively. Currents that are unidirectional
in time have a persistency of 100%.

**3   General characteristics of the Archipelago Sea currents**

Due to the temporal and spatial heterogeneity of the dataset, we divided it according to geographical location and season.
Geographically, we used four areas: 1. The southernmost part of AS, which has been shown to have a wider directional





distribution and lower current magnitudes than the other areas (e.g., Miettunen et al., 2024); 2. Central open areas of AS;
3. Narrow straits with fairways (e.g., Kanarik et al., 2018) and; 4. Northern areas, which are geographically located at the
Bothnian Sea, but have connection to AS through the narrow straits crossing the northern part of the AS.

The distribution into seasons was done based on variation in hydrography, winds, and ice conditions. The seasonal ther-
mocline is most prominent in July, August, and September, typically at depths of 15–20 m (Fig. 3). During other seasons,
temperature and salinity are vertically mostly homogeneous. The extent of ice in the Archipelago Sea is typically highest in
February and March. Wind conditions also show a large seasonal variation. Wind storms are frequent between October and
March, while the warm season between April and September is typically calmer (Laurila et al., 2021a). Also, the wind direction
has seasonal variation, e.g. NNW winds are more typical during warm seasons (Fig. 4). Based on these, we decided to use the
following division into seasons: Winter – January to March (JFM); Spring – April to June (AMJ); Summer – July to September
(JAS); and Autumn – October to December (OND).

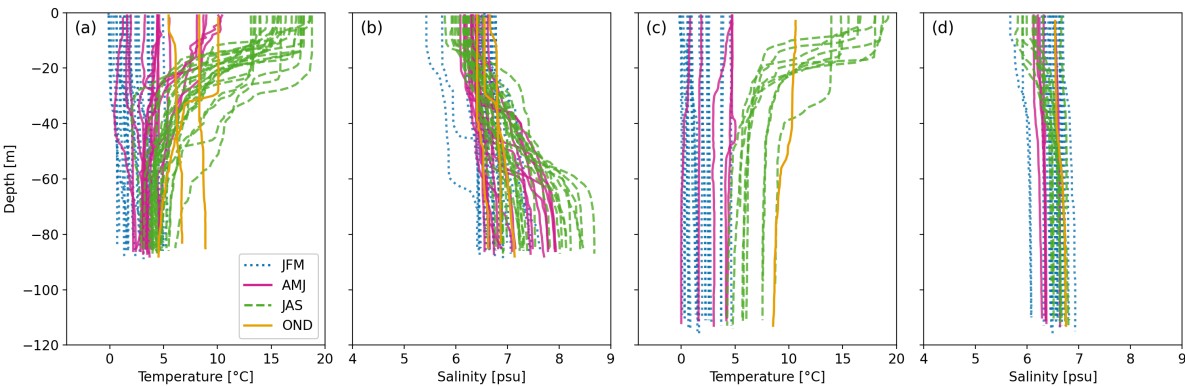

**Figure 3.** CTD profile measurements from 1991–2021 from IU7 (a,b) and IU6 (c,d) stations in the southern Archipelago Sea (locations
marked in Fig. 1). There are a total of 67 profiles from IU7 and 35 profiles from IU6. Colours represent different seasons.

**3.1    Currents in the southern edge of the archipelago**

Utö and Fölskär measurements represent the southernmost AS where the occasional halocline and seasonal thermocline divide
the water into two or three layers (see station IU7 in Fig. 3), of which the topmost, surface layer, is directly affected by winds.
The mean magnitudes calculated from the entire water column were around 6-7 $\mathrm{cms}^{-1}$ with higher values measured at the
Utö station than in Fölskär. The maximum magnitudes reached 58 $\mathrm{cms}^{-1}$ in Utö (surface layer) and 53 $\mathrm{cms}^{-1}$ in Fölskär
(intermediate layer[1], above halocline). Halocline in this area has a large seasonal and interannual variation in its occurrence
and depth, as shown, e.g., by Laakso et al. (2018). The presence of the halocline during our current measurement campaigns
can be observed from the change in flow directions in different layers of the water column from the two southernmost stations.

---

[1]Note that at Fölskär station, surface measurements were only available from mid June 2006 to end of the August at Fölskär station (Sec 2.1.1). Surface
measurements were not available at the time of the maximum measured current magnitude.





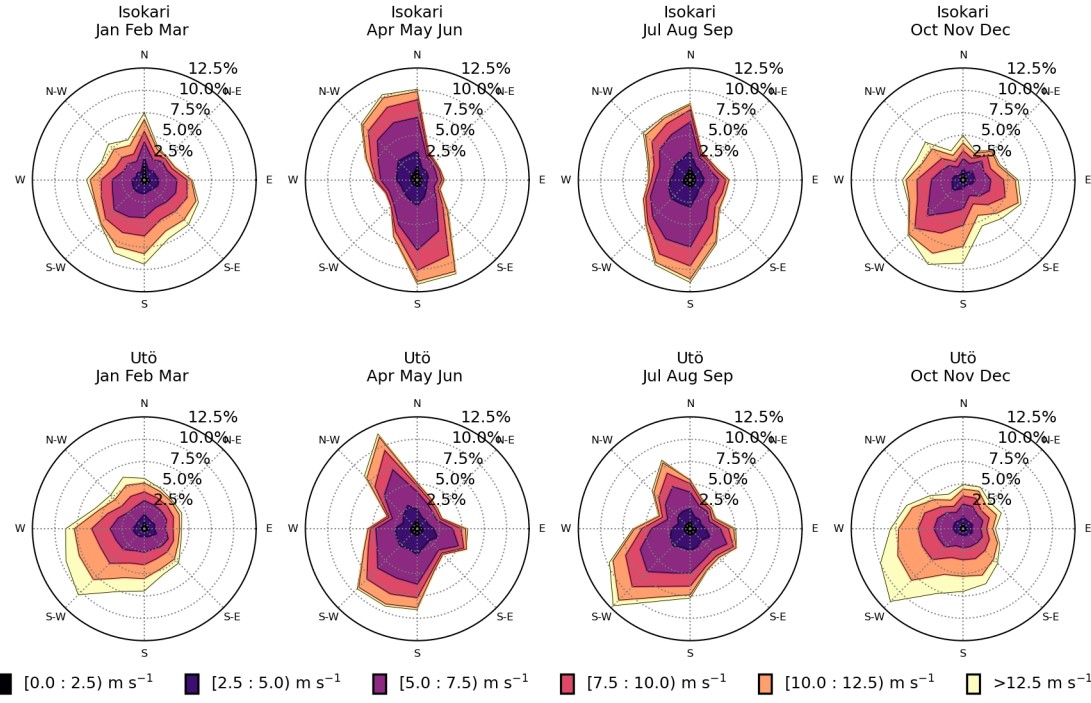

**Figure 4.** Seasonal difference of Isokari (upper) and Utö winds (lower). Directions show where the wind is coming from.

The prominent current direction in the surface layer was towards NW with persistency (P) of 38% in the westernmost strait (Fölskär, Fig. 5) with a mean magnitude of 7 cms$^{-1}$ and with P of 38%, and towards W at the easternmost station (Utö, Fig. 5)

with mean magnitude of 8 cms$^{-1}$ and P of 23%. At the bottommost layer, separated from the rest of the water column by the halocline, the flow was mostly steered by the bottom topography and directed towards the SSE (P of 24%) at the Fölskär station and toward the NW at the Utö station where the P was highest (49%) in the area. The mean magnitudes at the bottom varied between 4 and 6 cms$^{-1}$ with lower values measured in Fölskär. Even at the bottom, magnitudes up to 21 cms$^{-1}$ (Fölskär) and 38 cms$^{-1}$ (Utö) were measured.

The current directions in the intermediate layer, between seasonal thermocline and halocline, had a large spread and lowest persistencies (P). At Utö station, a large part of the currents were directed toward W/WSW with P of 17%. At Fölskär station, flow in the intermediate layer was evenly distributed between NW and SE with P of 19%, whereas at the surface the most frequent directions were towards NW than SE. Note that a large part of the simultaneous surface measurements were missing in Fölskär, making this layer less comparable to other layers. The mean current speed in this intermediate layer was 7 cms$^{-1}$

at both stations.





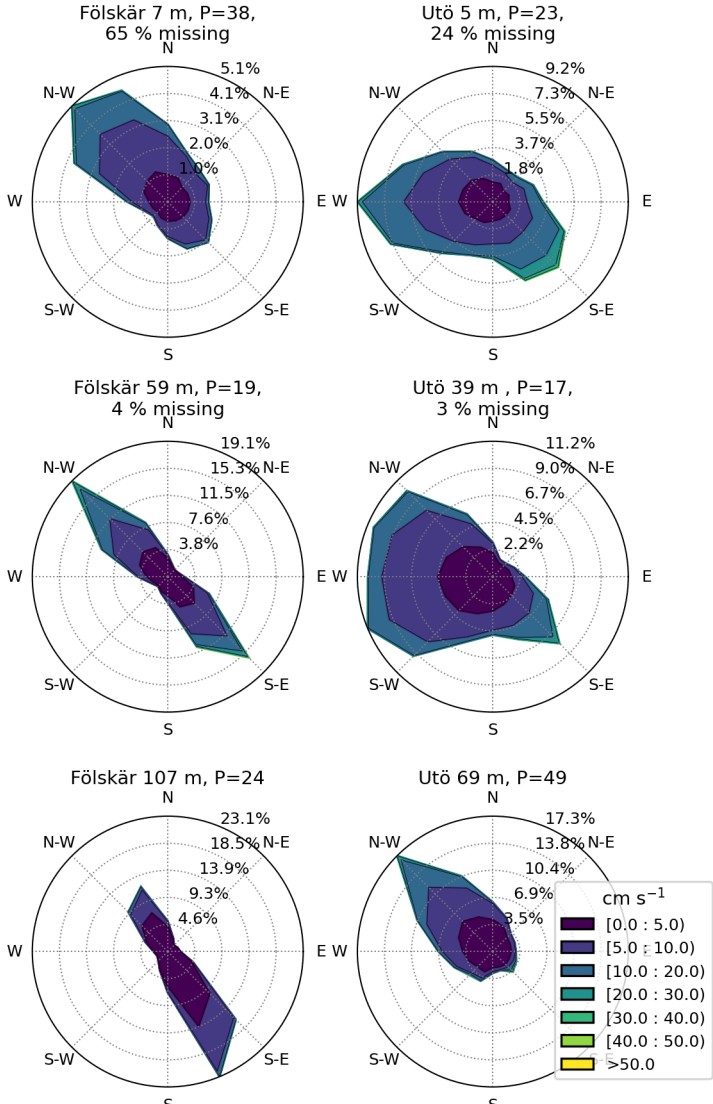

**Figure 5.** Current roses from the southern stations. The upper panels show the measurements closest to the surface, the middle panels currents on top of the halocline, and the lower panels the measurements closest to the sea floor (station depths in Table 2) with corresponding persistency (P, %). The percentage of missing data during the measurement period is shown for layers with significant amount of missing data. Directions show where the current is flowing to. For information of missing data, see Sect. 2.1.1.

## 3.2 Currents inside the archipelago

Measurements in the central areas of the AS were made in three different locations: Gustaf Dalen, Söderkobb, and Kihti. This central region is relatively open compared to other areas of the AS. However, the area is considerably more open in the NW-SE





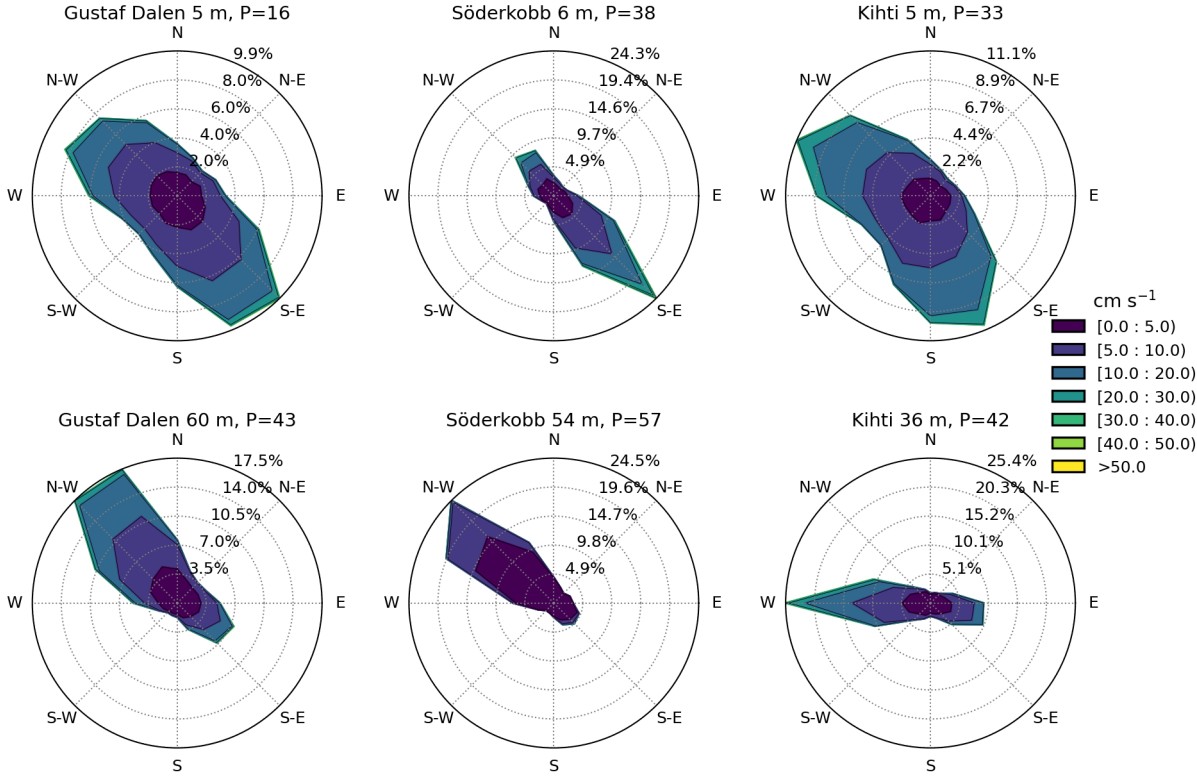

**Figure 6.** Current roses at the central stations. The upper row shows the measurements closest to the surface and the lower row the measurements closest to the sea floor (approximately 5 m from the sea floor, Table 2) with corresponding persistency (P, %). Directions show where the current is flowing to.

direction than in the SW-NE direction as there are small islets on both sides of the deep straits crossing the region. Due to this

geometry, the main flow direction of the currents at the surface is also toward NW or SE (Fig. 6 upper panels). The persistency at the surface layer was lowest at Gustaf Dalen (P of 16%), located at the crossing of the two deeper straits (Fig. 1), and about twice the value in surrounding Sörderkobb (P of 38%) and Kihti (P of 33%) measurement stations. The magnitudes in the surface layer varied between 8 cms$^{-1}$ and 9 cms$^{-1}$ with a higher value measured in Kihti.

      At the bottom, the bathymetry (Fig. 1) steers the currents along the deep straits (Fig. 6 lower panels). At the Kihti station, the

bottom currents were more frequently towards the west, with occasional flow to the east with a mean magnitude of 7 cms$^{-1}$ and P of 42%. At the Gustaf Dalen and Söderkobb stations the main direction was towards the NW along the channel at the bottom with corresponding mean magnitudes of 7 cms$^{-1}$ and 3 cms$^{-1}$ and P of 43% and 57%. The mean magnitude within the entire water column varied between 4 cms$^{-1}$ (Söderkobb) and 7 cms$^{-1}$ (Kihti and Gustaf Dalen). The maximum magnitudes were measured in the surface layer and ranged from 40 cms$^{-1}$ at Söderkobb to 55 cms$^{-1}$ at Gustaf Dalen. The low velocities



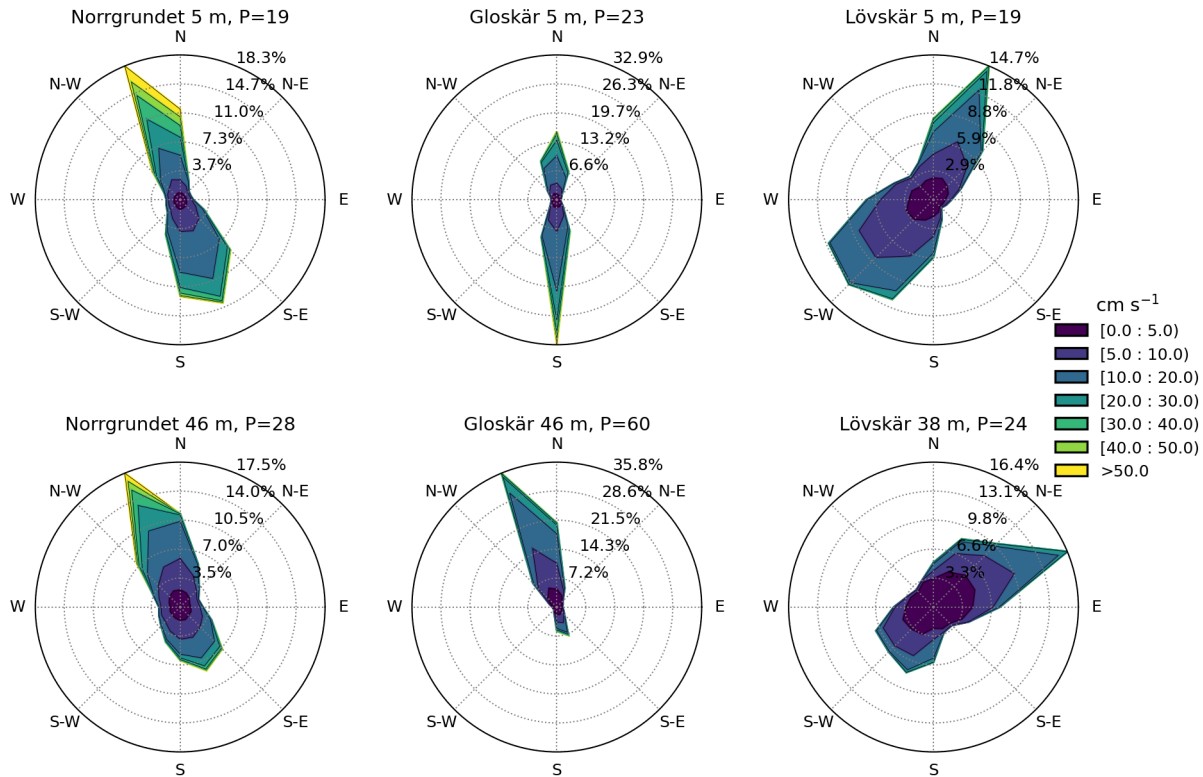

**Figure 7.** Current roses from the stations surrounded by the dense archipelago. The upper row shows the measurements closest to the surface and the lower row the measurements closest to the sea floor (approximately 5 m from the sea floor, Table 2) with corresponding persistency (P, %). Directions show where the current is flowing to.

and higher persistency at Söderkobb compared to the other two stations are explained by the seasonality of the measurements (Sect. 3.5).

## 3.3 Currents in the narrow straits inside the archipelago

Norrgrundet, Gloskär and Lövskär measurements were made in the narrow straits closely surrounded by the islands. Thus, the flow direction was more strongly orientated along the strait throughout the water column. Both of these straits are part of major
fairways through the AS where occasional strong currents can affect navigation of vessels, especially when perpendicular to the ship's course.

Norrgrundet and Gloskär stations were located in the same north–south aligned strait, which caused the flow to be strictly aligned towards NNW or SSE at Norrgrundet with 19% persistency (P) and toward N or S at Gloskär with P of 23%(Fig. 7). At the bottom layer, flow was most frequently towards the NNW at both station, however the P at the bottom most measurement
depth at the Gloskär was around twice (60%) the one of Norrgrundet (28%). This is explained by the bathymetry of the area



where Gloskär station was surrounded with shallower region also at south (Fig. 8 left pannel). These stations have almost double the mean magnitudes compared to other measurement locations in the AS, with the mean magnitude of the entire water column being 14 cms$^{-1}$ at Norrgrundet and 11 cms$^{-1}$ at Gloskär. The maximum measured magnitudes here reach up to 115 cms$^{-1}$ at Norrgrundet and 80 cms$^{-1}$ at Gloskär. The flow is slightly stronger in the surface layer with mean magnitudes of 14

and 17 cms$^{-1}$ compared to the bottom layer with 10 and 11 cms$^{-1}$, with higher values from Norrgrundet.

The narrowness and length of the strait strengthen currents along the flow direction, so that southward travelling currents are strongest at the southern edge of the strait and vice versa, as seen from simultaneous measurements of currents in Norrgrundet and Gloskär (Fig. 8). The distance between the stations is around 28 km (15 nautical miles) and within this distance, the current magnitude increases to around double in the northern end of the channel with northward currents. For southward currents, the

increase in magnitude along the channel is slightly less. Norrgundet measurements are located at the very end of the strait, thus having a longer overall fetch along the channel than the Gloskär site.

Lövskär station is located at the end of a much shorter strait than the one where Norrgrundet and Gloskär are located. Lövskär measurements were carried out in the cross section of two shallow straits and thus also in the most open area at that location (Fig. 1). Flow in Lövskär was mainly toward SW or NE, however, bathymetry and geometry steered currents at the surface

more toward NNE with a mean magnitude of 9 cms$^{-1}$ and P of 19%, and at the bottom towards NEE with a mean magnitude of 6 cms$^{-1}$ and P of 24% (Fig. 7, rightmost panels). NE currents have been shown to be mostly caused by south-easterly winds whereas SW currents were driven by northerly winds (Kanarik et al., 2018). The maximum measured current velocity at this station was 49 cms$^{-1}$ with a mean magnitude throughout the water column being 7 cms$^{-1}$.

### 3.4 Currents in the northern edge of the archipelago (in the southern Bothnian Sea).

The northernmost measurements (Norrskäret and Norra Bredan) were located at the northern ends of the westernmost and central straits coming out of the AS (Fig. 1). The mean magnitude of the currents measured throughout the water column in Norrskäret was 4 cms$^{-1}$ and 10 cms$^{-1}$ in Norra Bredan. The large difference between these areas is partly explained by the seasonality of these measurements, which are discussed in more detail later in Section 3.5.

The currents in Norrskäret were most frequently toward the WSW at the surface (Fig. 9, Norrskäret) with a mean magnitude

of 8 cms$^{-1}$ and P of 40%, while there was no clear main direction of the flow at lower depths with mean magnitudes of 3 cms$^{-1}$ and P of 15%. Overall, flow at the bottom in Norrskäret was more towards northward directions than toward southward. The directionality was quite different at the northeastern station, Norra Bredan. There, the flow was most frequently towards the south at the surface with a mean magnitude of 9 cms$^{-1}$ and P of 41%. Norra Bredan was located in the centre on narrow strait that strongly steered currents at the bottom layer, and thus the flow was mainly along the canyon (NNW/SSE) with P of

29%. The currents in the bottom layer in Norra Bredan were often stronger than at the surface with the mean magnitude in the bottom layer being 12 cms$^{-1}$. The strongest currents in this bottom layer reached 62 cms$^{-1}$ and were toward the SSE. The highest value measured in the surface layer of these areas was 50 cms$^{-1}$ at Norrskäret and 52 cms$^{-1}$ at Norra Bredan.





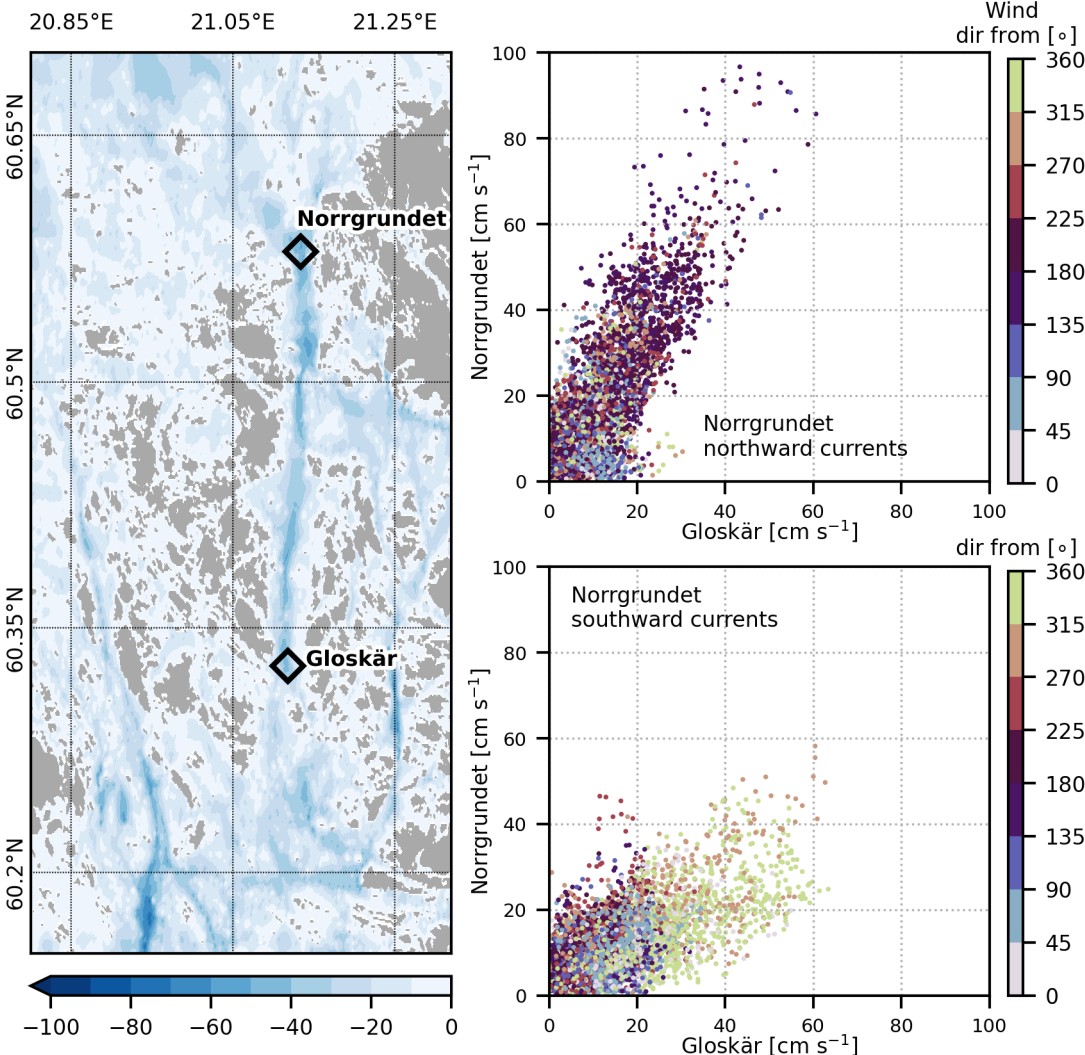

**Figure 8.** Bathymetry of the narrow north-eastern strait (left panel) and relation between simultaneous measurements (from 20 May to 3 Dec 2021) of Norrgrundet and Gloskär current magnitudes at 5 m depth, the uppermost measurement layer (right panels). Currents are divided to northward directed currents (upper right panel) and southward currents (lower right panel) according to the measured direction in Norrgrundet. The colour bar represents the direction of simultaneous winds measured at Isokari AWS.

## 3.5 Seasonal statistics of the current magnitudes

The maximum current speeds varied greatly between different measurement locations around the AS. The strongest currents were measured in the narrow straits in the northern part of the AS (Norrgrundet and Gloskär) and the weakest in the more central regions of the AS. The lowest maximum magnitude of 40 $\mathrm{cm s}^{-1}$ was measured in Söderkobb and the highest magnitude of





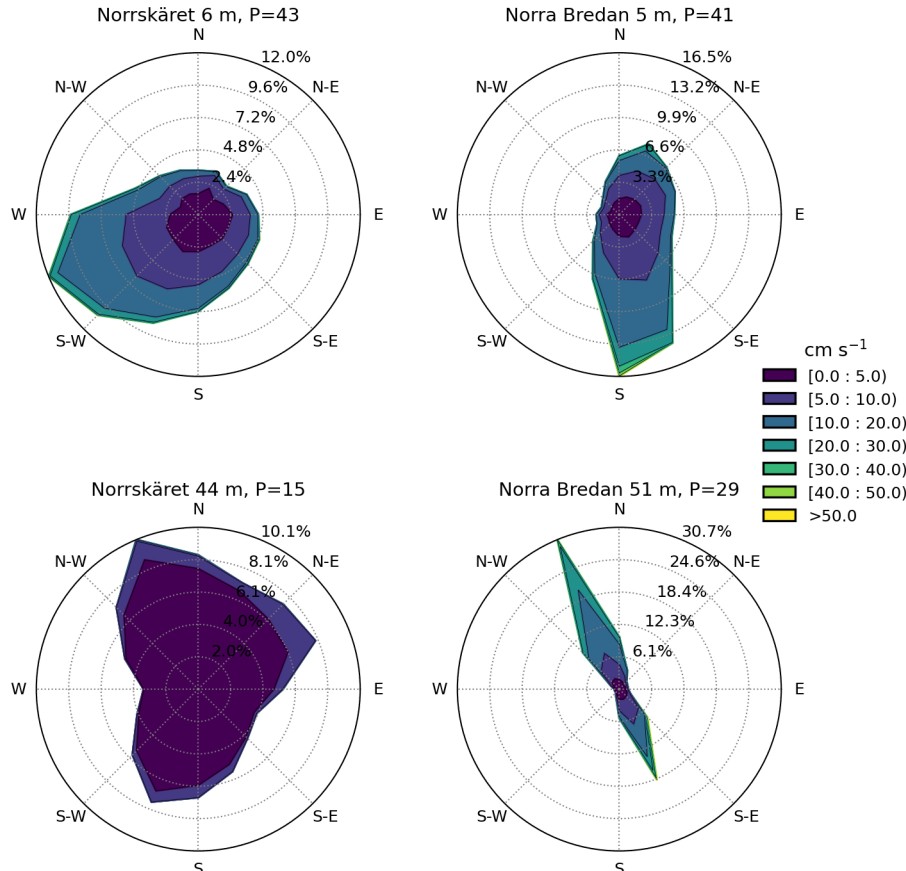

**Figure 9.** Current roses from the northern stations. The upper row shows the measurements closest to the surface and the lower row the measurements closest to the sea floor (approximately 5 m from the sea floor, Table 2) with corresponding persistency (P, %).

$115 \mathrm{~cms}^{-1}$ in Norrgrundet. This high magnitudes occurred rarely as 99% of the measurements at each measurement station were typically below half of the maxima (between 18 and $64 \mathrm{~cms}^{-1}$) and the mean current speeds varied between different regions from $4 \mathrm{~cms}^{-1}$ (Norrskäret) to $14 \mathrm{~cms}^{-1}$ (Norrgrundet).

Currents can have large seasonal and year-to-year variations due to variability in the meteorological and sea ice conditions, and stratification. Thus, the magnitudes of the currents at each station with different measurement periods are not directly comparable (Table 1, Fig. 2). To get an overview of spatial and temporal differences, seasonal statistics for each station were calculated over the whole water column (Fig. 10). In the analysis, we focus on the measurements in the southern and central parts of the AS.

The seasonal means of the current magnitudes (Fig. 10) are quite uniform within each season throughout the AS. In most of the AS stations, measured currents were weakest during spring when mean magnitudes varied between 4 and $6 \mathrm{~cms}^{-1}$. The currents were strongest in winter and autumn with mean magnitudes between 7 and $9 \mathrm{~cms}^{-1}$ in winter and $6$–$8 \mathrm{~cms}^{-1}$ in





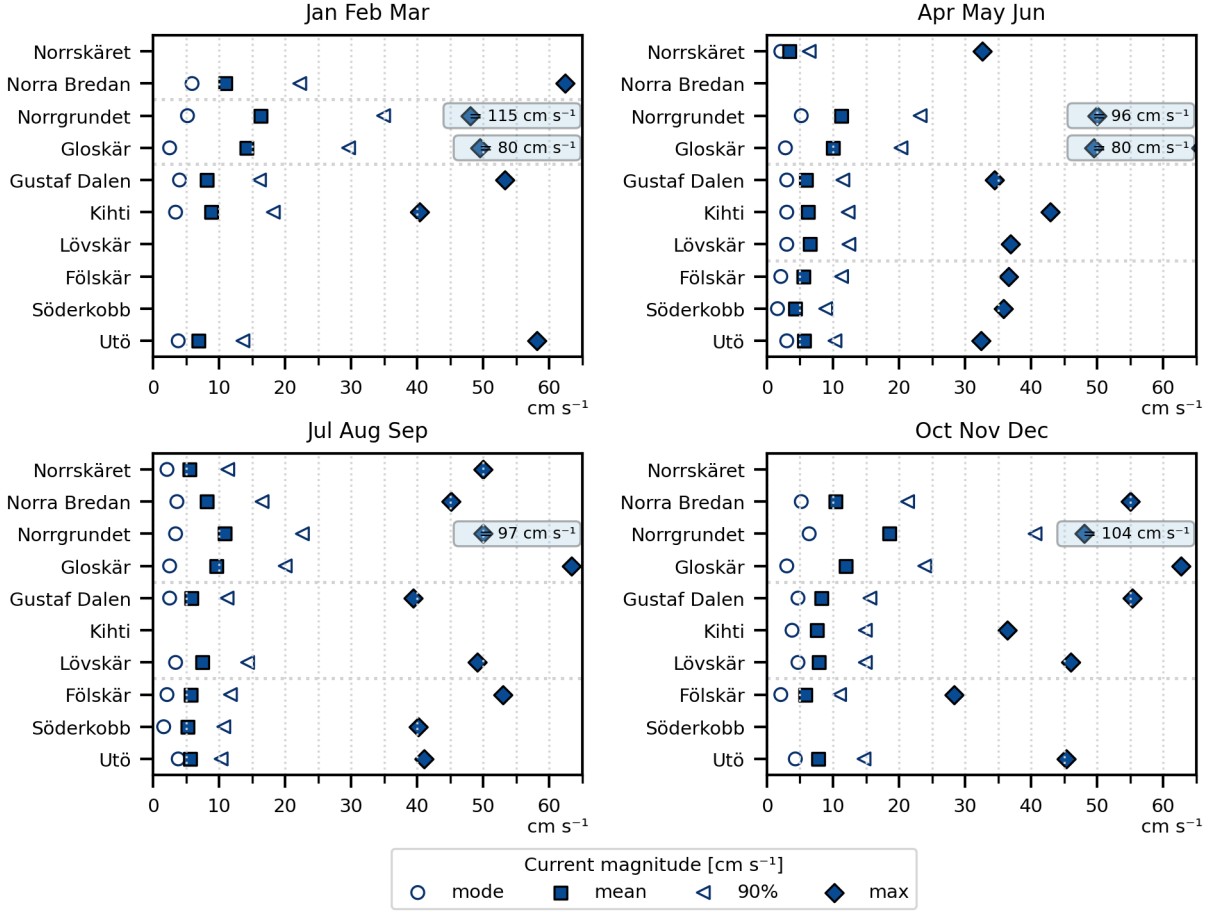

**Figure 10.** Seasonal statistics of current magnitudes over all depth bins. The value 90% represents the 90th percentile. The maximum values in the Norrgrundet and Gloskär stations exceeded the figure axis and were thus included in a separate box in the figure.

autumn. The mean magnitudes in summer ranged between 5–7 $\mathrm{cms}^{-1}$. Note that the most frequently measured values (mode) are below the mean value. The variations between different stations were larger when the maximum velocities were examined.

Extreme cases were typically caused by single storms, and the occurrence of these cases can vary largely in our heterogeneous dataset. We present the 90th percentile of the current magnitudes together with the measured maxima firstly to represent the rarity of the extremes and secondly to give an overview of the data distribution together with mode (Figure 10).

The largest variation in the current magnitudes between the measurement stations occurred during the winter season, when the winds are typically strong (Fig. 4) but occasionally the ice cover hinders the local winds from inducing currents. There are

also the fewest measurements from winter of all four seasons. Winter poses challenges to the ADCP measurements and typically some data had to also be discarded due to bad data quality in the Norrgrundet, Gustaf Dalen and Utö stations (Sect. 2.1.1).



Statistics are calculated only from good quality data, and thus current magnitudes in the winter represent mainly conditions during ice-free periods.

The mean and maximum magnitudes of the Norrgrundet and Gloskär currents are almost twice the values measured in other areas of the AS (Sect. 3.3 and Fig. 10). The strongest currents in the area were measured at the Norrgrundet station. The mean magnitudes for all seasons varied between 11 and 19 $\mathrm{cms}^{-1}$, with the weakest magnitudes always measured in summer and the strongest in autumn. One tenth of the measurements (90th percentile) exceeded the magnitudes of 23–41 $\mathrm{cms}^{-1}$ and the maximum values during the measurement period of around 30 months varied from 96 up to 115 $\mathrm{cms}^{-1}$, which is around 50 $\mathrm{cms}^{-1}$ higher than the maxima measured at the other stations outside this strait. The currents measured at Gloskär, in the slightly more central area of this strait, were slightly weaker and had a different distribution compared to those in Norrgrundet. The most frequent current magnitude (mode) was weaker than or in the same order of magnitude as in other stations in the AS. However, the mean, 90%, and maximum values were clearly higher than at the other stations outside of this strait. This means that strong currents occur more frequently in this strait than in the other measurement stations, although most of the currents at this station are of the same order of magnitude as at other locations in AS with a mode of around 3 $\mathrm{cms}^{-1}$. The mean magnitudes ranged from 10 $\mathrm{cms}^{-1}$ to 14 $\mathrm{cms}^{-1}$ and 10% of the values exceeded 20–30 $\mathrm{cms}^{-1}$ with the weakest statistics always measured in spring and summer and the strongest in the winter season. The seasonal maximums in Gloskär ranged between 63 and 80 $\mathrm{cms}^{-1}$.

During seasonal thermocline, measured currents in both Norrgrundet and Gloskär were significantly weaker in the bottom layer than in the surface (Fig. 11): the mean magnitude in the bottommost layer was around 7 $\mathrm{cms}^{-1}$ in Norrgrundet and 9 $\mathrm{cms}^{-1}$ in Gloskär[2]. The flow direction of the lower layer during stratification was mainly towards the north (Fig. 11 right panel), as also seen in Miettunen et al. (2024). The persistency was only slightly stronger in Norrgrundet during seasonal thermoclune (P of 32%) than the overall P in the bottom layer (28%), but at Gloskär the P increased to 84% due to the stratification compared to the overall 60%. These low current magnitudes below the thermocline during the summer season caused the mean current speeds (Fig. 10) to be more similar to the mean values of the other AS measurement stations outside of the strait. However, when we compared only surface layer statistics at Norrgrundet and Gloskär (not shown), current magnitudes in the summer followed the same trend as in other seasons of being almost twice the magnitude measured in other regions. Although currents below the thermocline were noticeably weaker than in the surface, there were events where the current magnitude in the bottom layer reached magnitude up 30 $\mathrm{cms}^{-1}$ at Norrgrundet and 38 $\mathrm{cms}^{-1}$ at Gloskär.

## 4   Drivers of strong currents

The Archipelago Sea serves as a route for maritime transport and is crossed by several fairways. Providing information on strong currents is important for maritime safety in this area, and therefore understanding the drivers of these currents is neces-

---

[2]These values were calculated by selecting times when current measurements showed clear stratification of the water column by having visible changes of magnitudes and/or flow direction (Fig. 11 before Sept 28). At Norrgrundet this meant time periods of 6 Sept to 28 Sept 2016, 5 June to 5 Sept 2017 and 17 June to 18 Aug 2021. In Gloskar, measurements from 17 June to 20 September 2021 and from 22 April to 2 June 2022 were used.



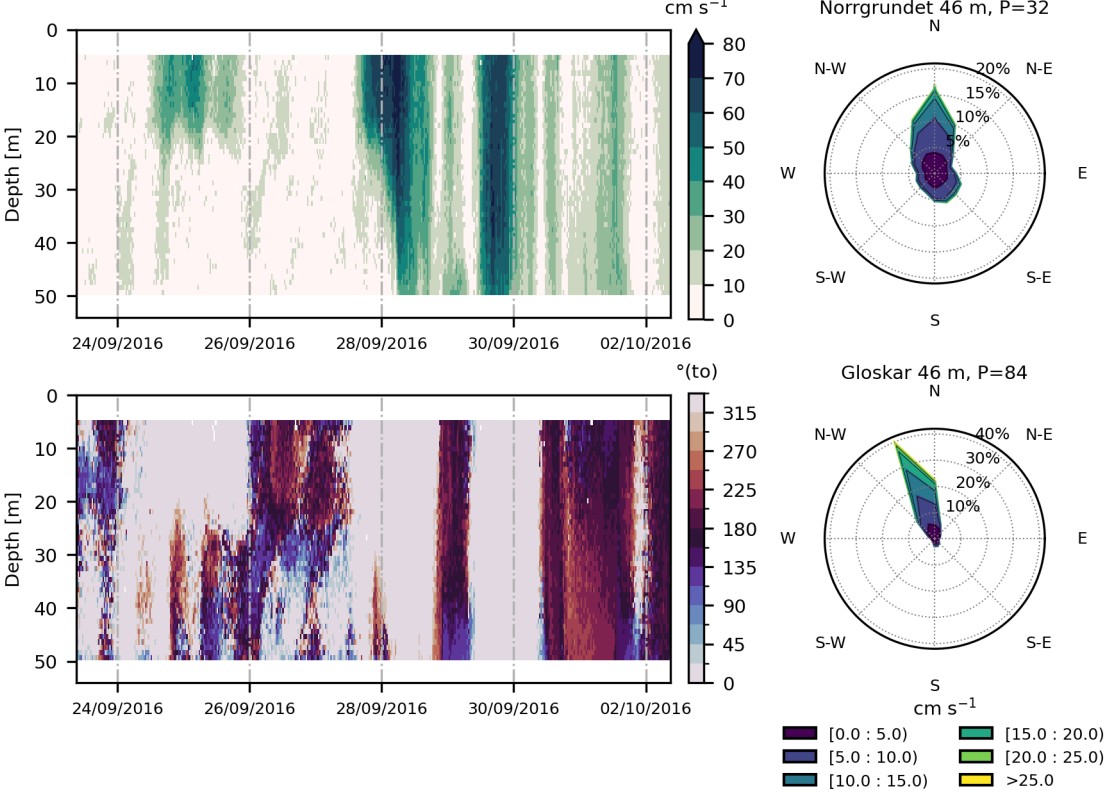

**Figure 11.** Left panels: Depth profiles of current magnitudes and directions at the Norrgrundet station before and after seasonal overturning on Sept 28, 2016. The thermocline can be seen at a depth of 20–30 m. Right panels: current roses in Norrgrundet and Gloskär at approximately 46 m depth during the stratified measurement periods with corresponding persistency (P, %). Norrgrundet data covers the time periods from 6 Sept to 28 Sept 2016, 5 June to 5 Sept 2017 and 17 June to 18 Aug 2021. Gloskär data covers the time periods from 17 June to 20 September 2021 and 22 April to 2 June 2022.

sary. We will focus our analysis on the long and narrow strait in the north-eastern AS (Sect. 3.3 and Fig. 8). The Norrgrundet station at the northern end of the channel had the highest current magnitudes of the entire measurement data set (Sect. 3.5). There were all together 224 events during c. 30 months of measurement time series, where the current speed exceeded 40

$\mathrm{cms^{-1}}$. The median and maximum duration of these events were 5.5 h and 35 h, respectively, which constituted almost 8% of the measurements. Most of these strong current events (c. 75%) aligned with the wind direction, while about one-quarter opposed the wind direction for the majority of the event's duration.

Strong current events are often the result of the combined effect of wind and differences in sea level. These forcings can work in the same direction, supporting each other, or in different directions, opposing each other. The complicated geometry

of the AS region causes the growth of currents driven by local winds to be highly sensitive to the wind direction. Even a





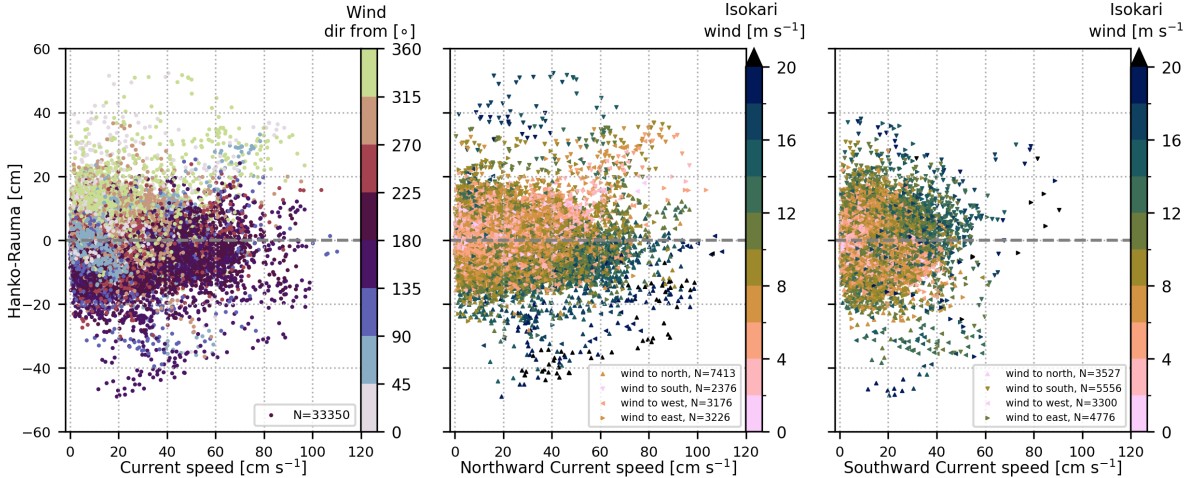

**Figure 12.** Norrgrundet current magnitudes against sea surface height difference between Hanko and Rauma. Left panels colour shows the simultaneous wind direction and middle and right panel the corresponding wind magnitude. For middle and right panels, the data is divided by the main current directions: middle panel has northward current events and right panel has southward events. In middle and right panels, the corresponding main wind direction is marked by marker arrow direction. Positive sea level difference drives currents northward and vice versa. Northward direction is defined as $> 315°$ and $\leq 45°$ and the other three main compass directions are defined correspondingly. N indicates the number of different data points in each scatter. Colourmap used in middle and right panels are from Crameri (2021).

slight shift in the wind direction can cease the growth of the currents. This pronounced sensitivity to wind direction for AS currents has previously been demonstrated in the narrow Lövskär cross section (Kanarik et al., 2018). In the north-south aligned strait studied here, the strongest currents typically occur with strong northward (Fig. 12, central panel) or southward (Fig. 12, right panel) winds, provided that the opposing sea level difference across the AS is not significant. When winds are weak or
misaligned with the strait's geometry, the sea level difference across the AS can become the dominant driver of strong currents (indicated by light colours with a large SSH difference in Fig. 12). Typically, the largest sea level differences across the AS coincide with strong opposing winds, meaning that the strongest currents rarely align with the highest sea level differences (Fig. 12 left panel).

      During the strongest current event, with a magnitude of $115 \ \mathrm{cms^{-1}}$ on 22 February 2017 (Fig. 13), both the wind and the
sea level forcing drove the northward flow, which was further amplified by changes in atmospheric pressure fields over the Bothnian Sea. In addition to external forcing, the narrowness of the channel further amplified the flow (Fig. 8). SSE winds with around $20 \ \mathrm{ms^{-1}}$ speed blew for several hours and the sea level was 9 cm higher in Föglö than in Rauma and Hanko (see the locations of the tide gauges in Fig. 1).

      The second strongest event, with a magnitude of $104 \ \mathrm{cms^{-1}}$ on 28 December 2016 (Fig. 14), followed the sea level gradient
over the AS and was formed while the wind was opposing the strong sea level gradient. During this event, a sea level difference of several tens of centimetres (up to 31 cm) between Hanko and Rauma counteracted the northerly winds, and this produced

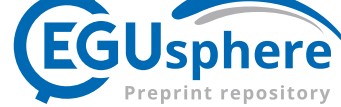

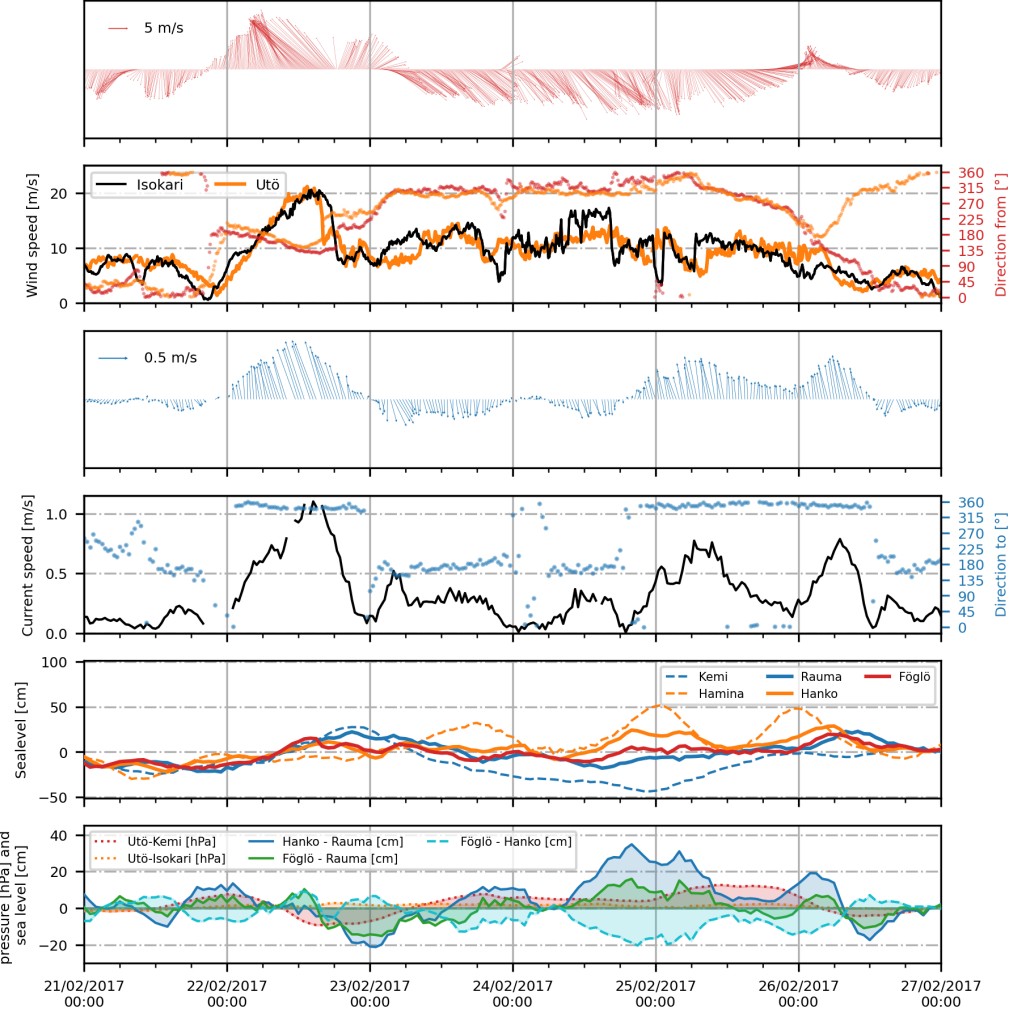

**Figure 13.** Time series the strongest current event in Norrgrundet (22 Feb 2017) followed by seiche induced wind opposing currents. Time series shows simultaneous winds, currents and sea levels together with the differences of sea level and atmospheric pressure in northern (Isokari and Rauma) and southern (Utö, Hanko and Föglö) edges of the Archipelago Sea (see the map in Fig. 1). Föglö represents sea levels in the southwestern AS and Hanko in the southeastern AS. Measurements in Kemi represent the conditions at the northern end of Gulf of Bothnia and measurements in Hamina represent the eastern Gulf of Finland. The first and second panels present wind speed and direction at Isokari, and third and fourth panels currents at the uppermost layer measured at Norrgrundet (5 m depth). Upward arrow indicates northward direction both for wind and current measurements. The fifth panel represents the sea level variation around the AS (solid lines) and at the end of the surrounding basins (dashed lines). The lowest panel shows the differences in sea level and atmospheric pressure between various stations around the AS.



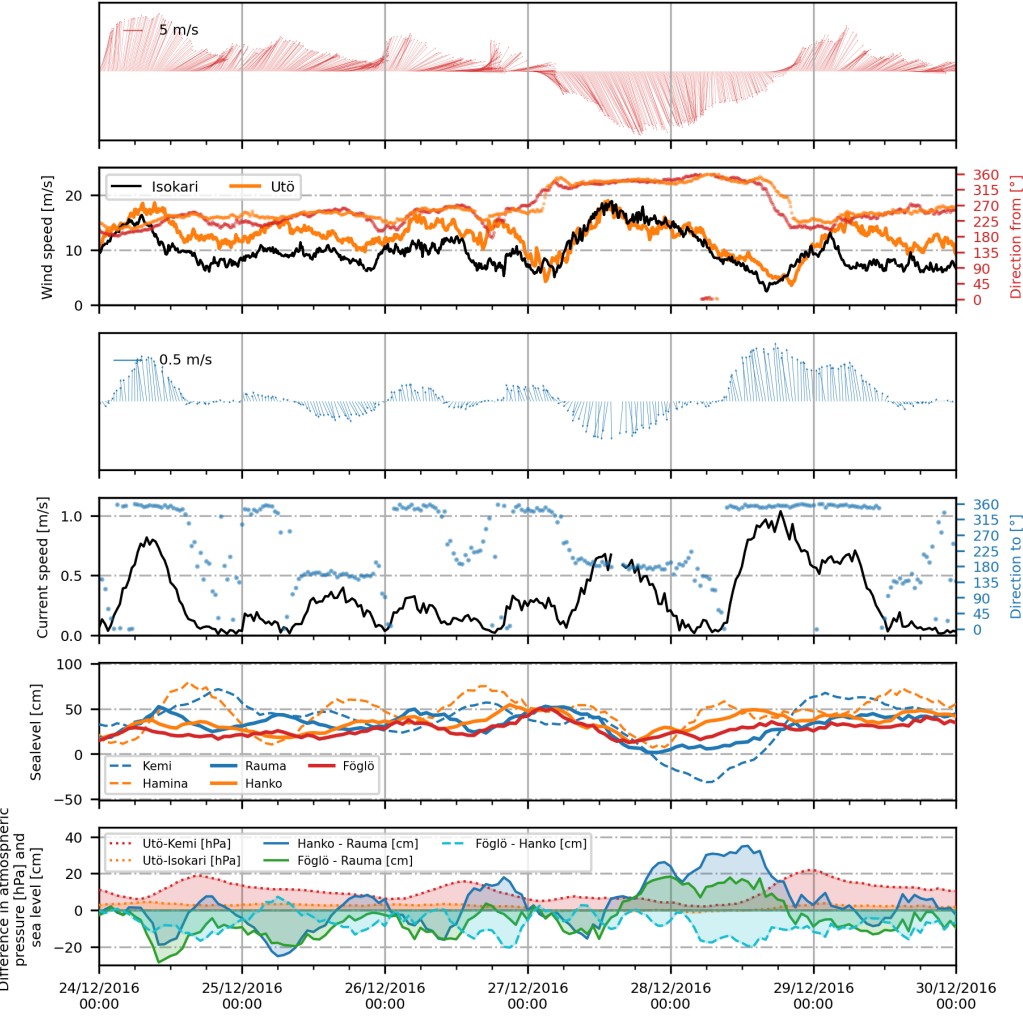

**Figure 14.** Time series the second strongest current event in Norrgrundet (29 Jan 2016) with simultaneous wind and sea level measurements together with sea level and atmospheric pressure differences from surrounding stations (Fig. 1). The first and second panels present wind speed and direction at Isokari (and Utö included with orange on the second panel), and third and fourth panel currents at the uppermost layer measured at Norrgrundet. The upward arrow indicates the northward direction both for wind and current measurements. The fifth panel represents the sea level variation around the AS (solid lines) and at the end of the surrounding basins (dashed lines). The lowest panel shows the differences in sea level and atmospheric pressure between various stations around the AS.

strong northward currents that flowed against the wind once the wind speed had degreased below $10 \text{ ms}^{-1}$. Note that the highest spike in current magnitude occurred when local winds shifted to blow along the strait. During this event, sea levels in the Gulf of Finland (Hamina measurements) had started to oscillate after the winds packed water to the end of the gulf on 345 24 December 2017 (seiche). Once sea level simultaneously fell at the Gulf of Bothnia (Kemi measurements) on 28 Dec, the




simultaneous seiche at the Gulf of Finland created such a strong gradient over the AS that it was able to oppose the weakened local wind. A similar event was observed between 24 and 26 February 2017, following the strongest measured currents (Fig. 13). These events are also well visible in comparison with current speeds, sea level differences, and wind magnitudes in Fig. 12. There, the light data points in the upper right corner of the central panel represent the numerous events where sea level

gradients drive strong northward currents. The phenomenon is not as visible in southward currents as the geometry of the strait further enhances the current speed, but is still present. Events related to the seiche in Gulf of Finland show better in comparison to Hanko and Rauma tide gauges than between Föglö and Rauma, which in general better represents the sea level gradient over the AS. During events with strong seiche, the sea level was also simultaneously higher in the SE corner (Hanko) of the AS compared to the SW corned (Föglö).

Mostly, the effect of opposing the sea level gradient to the wind direction only weakens the effect of local winds, whereas currents still follow the direction of the winds. During storm Toini on 11 January 2017 (Figure A2), the winds measured in Isokari were up to 22.5 $\mathrm{ms}^{-1}$ and stayed around 20 $\mathrm{ms}^{-1}$ for more than 18 hours. The maximum currents measures were 99 $\mathrm{cms}^{-1}$, being the third strongest event and had the longest lasting duration of currents over 40 $\mathrm{cms}^{-1}$ (35 h). These were the strongest winds measured during the 30 months of measurement at Norrgrundet station, and the storm resulted in record

significant wave height measurements by the Northern Baltic proper wave buoy (Björkqvist et al., 2017). However, during the event, the sea level difference quickly rose to oppose the currents being around 20 $\mathrm{cm}$ higher in Rauma than in the Föglö and Hanko stations in the south and finally the seiche in the Gulf of Finland made the sea level difference increase to 44 $\mathrm{cm}$ higher in Hanko than in Rauma. During the highest sea level difference, the current speeds had already decreased significantly. However, the currents did not begin to follow the sea level gradients before the wind speeds dropped below 10 $\mathrm{ms}^{-1}$.

During non-optimal wind conditions, sea level difference is the dominant factor driving the currents (as seen between 26–31 August 2021 in Fig. A1). Although easterly winds do not directly induce currents in the measured channel, they do affect sea level in more open sea areas, like at the northern edge of the AS, thus contributing to the lower sea levels measured at the Rauma station in this case.

It is notable that, in the AS, when the wind and the sea level gradient support the same flow direction, the wind speed does

not need to be very high to induce strong currents (dark green colours in Fig. 12). For example, on 18 Aug 2021 (Fig. A1), SSE wind only briefly reached the maximum of 16 $\mathrm{ms}^{-1}$ and stayed mainly only slightly above 10 $\mathrm{ms}^{-1}$ while simultaneously the water was high at the SE corner (represented by Hanko measurements) of the AS with sea level around 10 $\mathrm{cm}$ higher in Hanko than in Föglö and Rauma. Together, these resulted in up to 97 $\mathrm{cms}^{-1}$ current speeds at the Norrgrundet station, which is the fourth highest magnitude measured at this station.

## 4.1 Statistics of drivers

There are decades of atmospheric and sea level measurements from automatic weather stations (AWSs) and tide gauges along the Finnish coast (Sec 2.2). As our current measurements consist only of short datasets (Table 1), we extended our analysis to the statistics of the drivers identified as significant around the AS. We used a 30-year reference period from 1991 to 2020



to see the frequency of occurrence of high sea level differences and wind speeds. Continuous atmospheric pressure was only

available from 2007 onwards (Sect. 2.2).

### 4.1.1   Local winds

On average, the Utö AWS in the south measures slightly higher winds compared to the Isokari AWS in north. Three-hourly measurements from 1991 to 2020 show that the mean wind speed is 7.5 ms$^{-1}$ in Utö and 6.9 ms$^{-1}$ in Isokari, and 99%, respectively, being 18 ms$^{-1}$ and 16 ms$^{-1}$. The maximum measured wind speed during the 30-year reference period was 29

ms$^{-1}$ in Utö and 27 ms$^{-1}$ in Isokari, both measured on 14 Oct 2010. The seasonality of the local winds is briefly described in Sect. 3 and presented in Fig. 4 showing that the winds in the region are stronger in autumn (OND) and winter (JFM) than in spring (AMJ) and summer (JAS). For example, in Isokari, the 99% of the winds exceeded 14 ms$^{-1}$ during the spring and summer months (AMJ and JAS), while the same percentage was 18 ms$^{-1}$ in the autumn (OND) and 17 ms$^{-1}$ in the winter (JFM) months. Although record high winds were measured in autumn, up to 22 ms$^{-1}$ winds were measured in spring and 25

ms$^{-1}$ in summer at the Isokari station.

During our 30-month measurement period at Norrgrundet station, the distribution of the wind speeds followed rather closely the one from the climatological reference period, with 99% of winds exceeding 17 ms$^{-1}$ at both in Utö and Isokari. However, the maximum measured wind speed of 23 ms$^{-1}$ (storm Toini, Fig. A2), was still far from the maxima measured during the reference period.

### 4.1.2   Sea level differences

The 30-year reference period for tide gauge measurements compared to theoretical mean water shows that on average the water level is slightly higher in Rauma than in Hanko with a mean difference of 0.9 cm. However, extreme differences where the water level is higher in Hanko than in Rauma are slightly more frequent with 1% values greater than 22 cm and 0.1% greater than 39 cm, than those where the water is higher in Rauma than in Hanko with 1% values being greater than 19 cm and 0.1%

above 31 cm. Similar values are seen when looking at the sea level differences from a slightly different angle of the AS, as on average the sea level is 1.4 cm higher in Rauma than in Föglö with 98% of the difference varying between -17 to 14 cm (Föglö-Rauma).

Compared to the 30-year reference period, the Norrgrundet measurement time series reflected the typical distribution of sea level differences. The event on 29 December 2016, which featured maximum current speeds under opposing winds (Fig. 14),

was notably rare, with a sea level difference of 31 cm corresponding close to 0.1% occurrence frequency. Comparisons indicate that sea level differences across the AS can exceed those observed in our relatively short 30-month data set by several tens of centimetres. During the 30-year reference period, the maximum measured sea level difference was 71 cm between Hanko and Rauma and 59 cm between Föglö and Rauma.



### 4.1.3 Atmospheric pressure difference

To account for the inverse barometer effect, we examined measurements of atmospheric pressures over the AS, as strong atmospheric pressure differences can weaken or enhance the currents driven by the sea level difference. Sea level gradients caused by atmospheric pressure differences over the AS (measured between Utö AWS and Isokari AWS) typically vary only a few cm (or hPa) across the approximately 100 km distance between the two weather stations.

Simultaneous measurements, from 2007 to 2020, show that atmospheric pressure differences (Utö - Isokari) range between 415 -3 hPa and 4 hPa for 98% of the time. The maximum observed differences during the measurement period ranged from -6 hPa to 10 hPa. In the time series (Figs. 13 and 14) we also present the atmospheric pressure difference between Utö and Kemi, which have about 7 times the distance than between Utö and Isokari. Between Utö and Kemi, the pressure difference varies 98% of the time between -13 hPa and 20 hPa, with the maximum measured differences being -28 hPa and 29 hPa.

As the 1 hPa difference is equal to a 1 cm difference in sea level over the same distance, we see that the maximum tilt to the 420 sea surface caused by the atmospheric difference over the AS has been at most 10 cm and in 98% of the cases less than half of this.

During the 30-month measurement period at Norrgrundet, the maximum difference in atmospheric pressure between Isokari and Utö was 5 hPa in both directions. When looking at the strongest measured current events (Figs. 13 and 14), the atmospheric pressure difference has been at most 20 hPa over the entire Gulf of Bothnia (Utö-Kemi) and around 3 hPa over the AS (Utö- 425 Isokari).

## 5 Discussion

As this analysis is performed by combining measurements from numerous short campaigns, the measurement setups are not consistent with each other. Different measurements had varying depth cell size ensemble intervals and number of pings per ensemble, causing the standard deviation to vary between 0.7 and 1.75 cms$^{-1}$ (Table 2). Our measurements had multiple cases 430 where measurements were not successful due to non-optimal conditions at sea. The main scatterers in the right size range for 300 kHz ADCPs are zooplankton, which can be clearly seen from the analysis of the echo intensity of the signals at the site (Karvo, 2023). The signal quality appeared to be especially poor in springs after winters with substantial ice cover in the AS. However, this paper does not focus on the reasons behind these data gaps as they most likely have a strong biological background. Data were simply flagged as good on the basis of quality parameters provided by the instrument. As the prevailing 435 weather conditions have large seasonal as well as year-to-year variations, a direct detailed comparison between all the measurement sites should be made with caution. The maximum current speeds presented only reflect peaks during the measurement campaigns, and these depend on the type of storms in the area at that time. However, compared to previous measurements in the area (Virtaustutkimuksen neuvottelukunta, 1979; Ambjörn and Gidhagen, 1979; Kanarik et al., 2018), the measurement campaigns presented in this study had a larger coverage in both time and vertical extent (covering almost the entire water 440 column).



Our analysis is based on measurement campaigns that vary in time and location. We made seasonal analysis of the current magnitudes to make the measurements from different years and seasons more comparable. This analysis showed that narrow long straits significantly enhanced current speeds, being approximately twice the magnitude of the other areas, which mostly showed similar statistics to each other (e.g. mean speed around 8 $\mathrm{cms}^{-1}$). Although our measurements were only performed in one of the long north–south aligned straits, recent model analysis on circulation in the area (Miettunen et al., 2024) shows that the other straits show similar behaviour. In addition, one of the earlier measurement campaigns by Ambjörn and Gidhagen (1979) measured 91 $\mathrm{cms}^{-1}$ currents at the southern end of one of the other northern straits (north of Gustaf Dalen shown in Fig. 1). In general, our measurements show a very similar directional distribution of the currents as presented in the model simulations by Miettunen et al. (2024), although the measurements and model results cover different time periods. Combining information with measurements with suitable circulation modelling will in the future lead to a more comprehensive overview of the circulation dynamics also in those areas from which we lack measurements.

Our analysis was supported by weather station and tide gauge measurements from the AS and its neighbouring basins; the Gulf of Bothnia, Baltic proper, and Gulf of Finland (Fig. 1). These measurements represent the conditions in the surrounding basins and provide an overview of larger-scale phenomena that affect the dynamics of the AS. However, since weather and sea level measurements are from different areas around AS (Fig. 1), analysing the simultaneous effects of these drivers on currents, again at different locations, is challenging and prohibits a comprehensive overview of the different effects and their combinations.

The complicated configuration of islands and narrow channels of the AS affects the wind fields as well. The islands also have different topography and some are covered with dense forests. These can cause channelling effects on the wind field in narrow straits strengthening the winds (e.g., Chaudhari et al., 2023). This can then further enhance the current speeds. As there are no wind measurements available near the narrow straits, we could not quantify the effect. Considering a textbook example of surface currents being around 2–3% of wind speeds in the open sea, extreme wind events, with 30 $\mathrm{ms}^{-1}$ speeds, could induce a current speed of up to 60 to 90 $\mathrm{cms}^{-1}$ in idealised open sea conditions. The 2–3% estimate from the strongest measured winds of 23 $\mathrm{ms}^{-1}$ (storm Toini) during our measurement time series would give an estimate of 50 to 70 $\mathrm{cms}^{-1}$ currents. The measured maximum of the currents in Norrgrundet during the Toini storm was 99 $\mathrm{cms}^{-1}$, so the channelling effect has a significant contribution to the current speed in this narrow strait (Fig. 8). When considering the overall magnitudes of the surface current in the AS, where the mean magnitudes vary from 8 $\mathrm{cms}^{-1}$ in more open areas to around 14 $\mathrm{cms}^{-1}$ in narrow straits, the 2-3% estimate of mean wind speeds would largely overestimate the currents by giving the magnitude range from 14 to 22 $\mathrm{cms}^{-1}$.

Although atmospheric pressure fields and sea level differences work hand in hand in driving the sea currents, they could not be used directly together in these analyses. However, we can still get an overview of the sea-level gradients. Section 4.1 describes the long-term differences in sea level and atmospheric pressure over the AS and shows that although the atmospheric pressure differences can be quite large, the differences are clearly smaller than the largest differences observed in sea level (Sect. 4.1.2 and Fig. 12). Looking at the highest and lowest 1% of the differences over the AS, the atmospheric pressure difference between Utö and Isokari is approximately a quarter of the difference between Hanko and Rauma. To have more





exact comparison, sea level fields could be interpolated to the Utö and Isokari AWS locations using the surrounding three tide gauges. If we were to consider an idealised situation using these highest and lowest 1% of the sea level differences between Hanko and Rauma (around 20 cm) over a distance of around 200 km between the two stations, this tilt could drive geostrophic currents of around 8 cms$^{-1}$. The highest and lowest percentage of atmospheric pressure difference between Utö and Isokari (around 100 km apart) varied between 3 and 4 hPa. Considering again an idealised situation where 3 hPa air pressure difference would quickly change to equilibrium, this difference could drive around 1 cms$^{-1}$ currents. If we were to consider the rather unlikely condition of atmospheric pressure changing from one 1% extreme to another and thus changing the equilibrium by 7 hPa, this could drive currents up to 3 cms$^{-1}$. Currents possibly driven by atmospheric pressure changes are thus clearly weaker than the ones driven by overall sea level differences. Comparison to idealised current speeds over the area also shows that neither of these forcings grows strong enough to induce currents of the same magnitude as winds alone do. The maximum sea level difference during our Norrgrundet measurements was 31 cm between Hanko and Rauma, and under ideal conditions this could induce geostrophic currents of up to 12 cms$^{-1}$ while 23 ms$^{-1}$ winds could be estimated to include currents well above 50 cms$^{-1}$. Of course, these analyses are theoretical and not as such suitable for complex conditions of the AS.

As we for the first time have such a large data set of measured currents from the AS, we were able to catch extreme current speeds caused by sea level fluctuations in the basins surrounding the AS. Rantanen et al. (2024) has shown that sea level extremes in the Baltic Sea are typically linked to the clustering of many cyclones in the area. During the maximum measured currents and following the high wind opposing currents event in February 2017 (Fig. 13) there were four cyclones passing the Baltic Sea from different trajectories, allowing the sea level difference to first enhance currents speeds on Feb 24 and then grow strong enough to oppose the prevailing wind on February 28.

## 6  Conclusions

In this paper, we analysed and presented current measurement data sets from the Archipelago Sea (AS). We described the general features of the measured currents and also examined the drivers of the strong currents in the area.

Measurements in the AS can be divided into two types based on current magnitudes: more open sea areas with a mean surface layer magnitude of around 8 cms$^{-1}$ and narrow long straits with mean surface magnitudes of around 14 cms$^{-1}$. All of our data sets have measured currents of at least 40 cms$^{-1}$. In more open areas, maximum currents typically varied between 50–60 cms$^{-1}$ but up to 115 cms$^{-1}$ have been measured in narrow straits. However, these high values are still rare, as 90% of the measurement values were below 10 to 20 cm and 99% of the values were around half of the maxima.

The flow direction in the AS is bimodal as the geometry of the area restricts the flow. Due to the geometry, currents in the northern AS are strongly aligned to the north-south directions following the narrow channels crossing the AS. In central areas, such as Gustaf Dalen, currents are more widely spread with main flow directions to NE and SE, whereas in the southern most Utö stations the flow has significant westward component.

Our analysis shows that local winds are the main driver of strong surface currents in the AS and that these currents are very sensitive to changes in wind direction. However, the sea level difference over the AS has a large role in enhancing or opposing



the growth of the currents. Large sea level differences are formed mainly because of strong local winds, and the sea level
difference over the AS starts to prevent current growth after the wind has blown long enough.

For sea level gradient to dominate the flow direction, the local winds need to either be weak enough or to blow from non-optimal direction related to the open sea. Our analysis shows that currents caused by large sea level differences can grow very strong when there are large sea level differences in the surrounding basins, independent of the local winds. The most notable cases in which sea level dominates over local winds occur during strong seiches in the Gulf of Finland with simultaneous low
sea levels in the Gulf of Bothnia, both driven by larger-scale atmospheric phenomena over the entire Baltic Sea. In this paper, we have presented two events in which low sea level in the Gulf of Bothnia coexisted with strong seiches in the Gulf of Finland both resulting in very strong currents well above $70\,\mathrm{cms^{-1}}$ opposing the direction of the prevailing wind (24 February 2017 in Fig. 13 and on 27 Dec 2016 onward in Fig. 14). In the AS, even weaker winds, together with the supporting sea level gradient caused by oscillations in surrounding basins, can result in very strong currents in the narrow straits crossing the area (see event
on 18 August 2021, Fig. A1).

*Data availability.* Wind and sea level data are openly available from the FMI open data portal (https://en.ilmatieteenlaitos.fi/open-data). Bathymetry information is available through EMODnet Bathymetry (https://emodnet.ec.europa.eu/en/bathymetry). CTD measurements are available either though https://www.marinefinland.fi/en-US or through ICES data system (https://www.ices.dk). The ADCP data presented will be available through FMI data storage **(https://doi.org/XXXXXX/XXXX, Xxxxxxx et al.,2025)**



**Figure A1.** Time series of Isokari winds and currents from Norrgrundet and Gloskär with simultaneous wind and sea level measurements together with sealevel and atmospheric pressure differences from surrounding stations (Fig. 1). The first and second panels present wind speed and direction at Isokari (and Utö in orange on second panel), third and fourth panel currents at the uppermost layer measured at Norrgrundet and Gloskär. The upward arrow indicates the northward direction. The lowest panel shows the differences in sea level and atmospheric pressure around the AS.





**Figure A2.** Time series the third strongest event of Norrgrundet currents (storm Toini on 11 Jan 2017) with simultaneous wind and sea level measurements together with sea level and atmospheric pressure differences from surrounding stations (Fig. 1). The first and second panels present wind speed and direction at Isokari (and Utö in orange on second panel), third and fourth panel currents at the uppermost layer measured at Norrgrundet. The upward arrow indicates the northward direction for wind and current measurements. The fifth panel represents the sea level variation around the AS (solid lines) and at the end of the surrounding basins (dashed lines). The lowest panel shows the differences in sea level and atmospheric pressure fields around the AS.



*Author contributions.* The study was initiated by PA and LT. HK was responsible of the processing and analysing the ADCP datasets, and curating and presenting the data presented other data in this paper. EM performed the analyses of the hydrography of the area and figures 1 and 3. All authors took part in the extensive discussion of the results and provided expertise and incite to the conclusions from their area of expertise. HK and LT prepared the manuscript with contributions from all authors.

*Competing interests.* The contact author has declared that none of the authors has any competing interests.

*Acknowledgements.* This study would not have been possible without our colleagues at the FMI's (and previous FIMR's) Observation Services. Thank you for deployments, technical services and insights of the work at sea to Riikka Hietala, Tero Purokoski, Heini Jalli, Pekka Kosloff and to the others who have worked with us at the sea, and of course the whole crew of RV Aranda for flexible work with us. This work was supported by a grant from the Vilho, Yrjö and Kalle Väisälä Foundation of the Finnish Academy of Science and Letters, the Finnish Ministry of the Environment (Water Protection Programme 2019–2023), the European Union's European Maritime and Fisheries Fund, and
national Finnish Marine Research Infrastructure (FINMARI) consortium.



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
