# Peer review of "Currents and their Drivers in the Archipelago Sea: Insights from ADCP measurements"

_EGUsphere, 2025_

## Referee Comment (RC2)

**Review of "Observed currents in the Archipelago Sea"**

This manuscript presents a comprehensive observational study of currents in the Archipelago Sea (AS), an extremely complex coastal area in the Baltic Sea with myriad islands and narrow straits. The authors have compiled Acoustic Doppler Current Profiler (ADCP) measurements from 10 locations over roughly two decades of short campaigns - a valuable dataset that fills a longstanding gap in Baltic Sea observations.

The study is highly relevant to Ocean Science, addressing important questions about coastal circulation patterns and extreme currents in an archipelago environment. The manuscript's overall scientific quality is good: the ADCP measurements are well described with appropriate quality control, and the results are presented thoroughly with substantial interpretation. The findings are significant for both scientific understanding and practical applications, as the authors quantify typical current magnitudes and demonstrate the conditions for powerful currents. Notably, the paper identifies local wind as the primary driver of currents but also highlights how basin-scale seiches and local sea-level differences can generate strong flows when local winds are weak or unfavorably oriented. However, the authors' claims would be more convincing if they were better supported by the available observational data or complemented by modeling, which could help clarify the intervariability among parameters. Overall, the manuscript is well structured and written in generally clear language, although the clarity of the writing and the presentation of the figures could be further improved. It represents a useful and original contribution that fits well within the scope of Ocean Science.

**General comments**

**Data Gaps and Seasonal Bias:** Some deployments suffered from significant data gaps. For instance, at Fölskär only 35% (or to be corrected?) of near-surface data was obtained due to summertime acoustic issues (implied around lines *117–126*). The authors should discuss whether such gaps might bias the analysis of seasonal currents. Do missing summer measurements (e.g., June–July at Fölskär) affect the calculated mean speeds or directional distributions for that location? A brief comment on if and how these data gaps were handled would be useful.

**Combining Short Datasets:** The study compiles current measurements from multiple short-term campaigns across two decades. While this provides broad coverage, it also raises comparability concerns. The authors acknowledge differences in measurement setups and varying weather during each deployment. It would be helpful if they elaborated on if (and if yes) how they accounted for these differences when comparing sites. The paragraph about unifying all measurements into seasonal statistics (lines 270-275) is not convincing to make such analysis based on Fig 10. Perhaps indicating seasonal data completeness in Fig 10 would help with interpretation of statistics (e.g perhaps other stations with missing winter data could also encounter strong currents).

**Persistency Metric Usage:** The persistency (P) metric is defined in Section 2.3 (line 165–170) as a

measure of how consistently currents maintain one direction. However, P is mentioned only briefly in the text, mostly through unnecessary numerical statements without further explanation or discussion. It would strengthen the manuscript to report and interpret P values for the different regions. For example, how persistent are the flows in narrow straits versus open areas or southern vs northern AS? Reasons for low P (high or low frequency oscillations) including a short commentary on persistence (perhaps in Section 3 or 5) would give a quantitative sense of directional variability and complement the qualitative descriptions.

**Seiche- vs Wind-Driven Flows**: A major finding is that about 75% of strong current events align with local wind, while ~25% oppose it due to sea-level gradients (lines 319–322). This is an important result for understanding forcing mechanisms, however this is hard to see from the later analysis of Fig 12-14, therefore all of the conclusions and statements on these figures are a bit vague. Further the authors might consider expanding the discussion on how to predict or identify these sea-level-driven events as the navigation safety is mentioned multiple times. For instance, could a combination of low local wind (<10 m s$^{-1}$) and a large Gulf of Finland seiche be a practical indicator of an upcoming current reversal?

**Contextualizing Findings:** The discussion could better contextualize these results relative to previous knowledge and broader implications. The authors do compare their measurements to past campaigns and models (lines 443–451), noting similar behavior in other straits and agreement with model simulations. It would be worthwhile to explicitly highlight what is *new*. For instance, is this the first direct evidence of >100 cm s$^{-1}$ currents in the Archipelago Sea (confirming model predictions)? Do the results alter our understanding of AS circulation, or mainly reinforce existing model expectations? A few sentences synthesizing how this extensive dataset advances knowledge would be valuable. Additionally, since safety of navigation is a motivation (lines 225-226), the authors might comment on how their findings could be used by mariners or somehow integrated into warning systems in the future.

**Title could be refined:** it is accurate, but overly broad and generic. It doesn't reflect the depth of the analysis or the study's key contributions. A more descriptive and specific title would help the paper stand out and communicate its scientific focus better.

**General comment regarding figures:**

- Consider adding letters to figures instead of (lower right panel). See the"Figure content guidelines:" in journals' submission guidelines.
- Avoid crowded scatter plots, where the majority (?) data is overplot.

**Specific Comments:**

- **(line 5)**: *"lack of quality ensured measurements"* – This phrasing is odd. It should likely be **"lack of quality-assured measurements"**, since the authors mean there were no previously quality-controlled current observations.

- **(line 13)**: *"northern end of the long NE strait"* – The term "NE strait" is used without prior definition. Readers might not know *NE* means north-eastern here. Consider spelling out or referring to it as "the north-eastern strait" for clarity.
- **(line 14)**: *"surrounding basing"* – Typo, **"basing"** should be **"basins."** (e.g., "oscillations in the surrounding basins").
- **(line 15)**: *"non optimal direction related to the straits"* – Awkward phrasing. It would read better to use a hyphen in "non-optimal" and clarify "related to" as "relative to" or "with respect to."
- **(line 30)**: *"using a large network of light ships"* – Consider clarifying **"light ships"** if this is a historical term (lightships). For modern readers, a brief note that these were essentially floating lighthouses used for measurements might help, but this is minor.
- **(line 66)**: *"have been largely unpresented and published before this paper."* – This seems contradictory. Likely a mistake: perhaps **"unpresented and unpublished"** was intended.
- **(line 69)**: *"inner parts of the AS and concluded that the interactions are extremely complicated due to the heterogeneity of the area."* – Suggest adding a explanation if possible. It's understandable, but "extremely complicated" interactions could be elaborated in manuscript, since the reference is a report in non-English and very old one. Perhaps (if available) refer to some recent study which elaborates the "extreme complexity.
- **(line 76)**: *"even 15 m/s SW winds were not able to induce strong currents in this area"* – Perhaps specify **"in that area"** (Lövskär cross-section) to make it clear you are referring to the specific site studied by Kanarik et al. (2018), not the whole archipelago.
- **(Table 1)**: The table of ADCP moorings has some formatting issues.
  For example, the second entry **"SVM2S 25 Apr – 25 Aug 2006"** lacks a location name. Is this the Fölskär measurement (assuming from Fig 2.)? Please ensure each dataset is clearly named (or clearly grouped) in the table for cross-reference.
  Also, the note *'device changed on XXX'* should be included as a footnote or placed in an additional column, since the current column is intended specifically for the observation period.
- **(line 125): Devise** should perhaps be device.
- **(line 127):** without the technical understanding or description about the technology behind ADCP devices it is hard to understand what are the 'ensambles' and 'solutions' that ADCP is unable to calculate.
- **(line 131)**: The description of data return at Fölskär is a bit hard to parse. It says only 35% of near-surface measurements were successful above the thermocline. Though I can not conclude from Fig. 2, where the amount of "good data" seems to be significantly higher.
- **(line 133)**: *"vertical velocities and thus error velocities were exceptionally high at the bottommost 4 bins."* – Good that this was caught. The fix (removing bottom four bins) is fine, but maybe mention in the Table 2 or text (perhaps ref ID) which dataset instead of "Norrgrundet second deployment" this concerns, for clarity.
- **(line 150)**: *"Continuos simultaneous data of atmospheric pressure from multiple ASW stations…"* – Two issues here. **"Continuos"** is a typo; it should be **"Continuous."** Also, **"ASW stations"** appears to be a typo for **"AWS stations"** (Automatic Weather Stations). Please correct these to avoid confusion.

- **(line 163)**: *"This data is available…"* – Technically, **"data"** is plural. Perhaps say **"These data are available…"** (also at line 147). Consistent plural use would be preferred (e.g., "Data were collected…" rather than "data was").
- **(line 168)**: The formula for persistency (P) is presented. Ensure that all symbols ($\sum$, n, N, u_n, v_n) are properly defined in the text. Currently, *"where u and v are the eastward and northward velocity components…"* is given. Perhaps also clarify that *n* indexes each observation in the time series and *N* is the total number of observations, for completeness.
- **(line 172–176)**: The definition of four geographic sub-regions is useful. However, it might help to explicitly list which measurement sites belong to each region (perhaps in the caption of Figure 1 or in Section 3.1–3.4 headings or in Tables). Currently the reader must infer from Section 3.1 text which sites are "southern edge." A direct mapping (e g on Fig) would improve clarity.
- **(line 185)**: *"3.1 Currents in the southern edge of the archipelago"* – Consider capitalizing Archipelago Sea when referring to the region (consistency issue). Also "southern edge" vs "southern part" – just ensure consistency with earlier nameings.
- **(line 186)**: *"the occasional halocline and seasonal thermocline divide the water column"* – verb agreement: *"divide"* should be **"divides"**. Or rephrase the sentence for clarity.
- **(line 206)**: *"3.2 Currents inside the archipelago"* – Same naming issue as earlier 3.1 paragraph. Perhaps use **"central archipelago"** to align with the earlier description of Area 2.
- **(line 220)** "*higher persistency at Söderkobb compared to the other two stations are explained by the seasonality of the measurements (Sect. 3.5).*" raises the question of whether this higher persistence is actually due to missing observations during colder periods at Söderkobb, unlike the other stations. It would be helpful to elaborate on this point in Section 3.5 and discuss whether the observed persistence is a statistical artifact resulting from data gaps or a true physical feature. In a similar manner, it would be worthwhile to reassess the rest of the analysis to ensure that patterns attributed to physical processes are not instead influenced by data gaps or seasonal biases.
- **(line 313)**: *"reached magnitude up 30 cm s$^{-1}$ at Norrgrundet"* – Missing a word: **"up to 30 cm/s."**
- **(line 319)**: *"There were all together 224 events during c. 30 months…"* – **"all together"** should be **"altogether"** (one word) in this context. Also consider replacing *"c. 30"* with **"approximately 30"** for clarity (casual readers may not recognize "c." as "circa").
- *(line 327-333)*: It seems to be one of the key results. However it is hard to interpret from the Figure 12. As much as I can read from the figure it seems that the strongest northward currents (>100 cm/s) are mainly related to "winds to west" rather than "to north". Same for southward currents where "winds to east" could be related to strong currents. Perhaps some multi-linear regression or clustering technique could be used to make more solid estimates with interrelationships between current speeds, sealevel and wind characteristics. Currently these inter-relationships are a bit vague and almost impossible/controversial to see from the colorful scatter plots.
- **Figure 8.** Add Isokari AWS, if it fits to the region
- **Figure 12.** It is hard to interpret the relationships given in text from the figure. Consider better phrasing, additional annotation - or completely redoing the figure. Too much information makes the reading hard - and even controversial (see previous comment regarding lines 327-333).

Consider focusing only on the key data that effectively delivers the message, rather than including all measurements, which can overwhelm the reader. If, during revision, you find that all data points are indeed necessary, avoid plotting them all on top of each other, as this currently results in displaying only a partial segment at the end of the time series. Additionally, it would be clearer to use a consistent approach to wind direction within a single figure - either 'winds to' or 'winds from' - to avoid confusion, or justify the use of different notation.

**(line 334)**: *"During the strongest current event, with a magnitude of 115 cm s$^{-1}$ on 22 February 2017 (Fig. 13), both the wind and the sea level forcing drove the northward flow…"* – The phrase **"sea level forcing"** might be unclear. Maybe say **"sea level gradient"** or **"sea-level setup"** to be explicit.

- **(line 336)**: *"In addition to external forcing, the narrowness of the channel further amplified the flow (Fig. 8)."* – Here *"external forcing"* refers to wind and pressure presumably. This is fine, but perhaps specify: **"In addition to wind and sea-level pressure forcing…"** for clarity, since "external" could be interpreted as outside the strait.

- **(line 337)**: *"SSE winds with around 20 m/s speed blew for several hours and the sea level was 9 cm higher in Föglö than in Rauma and Hanko"* – It would be clearer as: **"SSE winds of ~20 m/s blew for several hours, and during this time the sea level at Föglö was about 9 cm higher than at Rauma and Hanko…"**. This explicitly states which location had the higher water level.

- **(line 339)**: *"The second strongest event, with a magnitude of 104 cm s$^{-1}$ on 28 December 2016 (Fig. 14), followed the sea level gradient over the AS and was formed while the wind was opposing the strong sea level gradient."* – There is potential confusion with dates (see next comment). Assuming 28 Dec 2016 is correct, the wording "followed the sea level gradient" could be rephrased as **"occurred in response to a sea-level gradient across the AS"** or **"was driven by a sea-level difference across the AS, with wind blowing in the opposite direction."**

- **(Fig. 14 caption)**: There is a **discrepancy in dates for the second strongest event**. The text says *28 December 2016*, but the Figure 14 caption caption refers to *"(29 Jan 2016)"* – which seems incorrect. **This is confusing and likely an error.** Please clarify the timeline of events:

- **(line 342)**: *"wind speed had degreased below 10 m s$^{-1}$."* – **"degreased"** is a typo; should be **"decreased."**

- **(line 344-346)**: *"winds packed water to the end of the gulf on 24 December 2017 (seiche). Once sea level simultaneously fell at the Gulf of Bothnia on 28 Dec, the simultaneous seiche at the Gulf of Finland created such a strong gradient over the AS that it was able to oppose the weakened local wind."* – The dates here are likely wrong given context. Assuming it should be late 2016. Consider rephrasing eg. "winds had piled up water at the head of the Gulf of Finland on 24 Dec 2016, initiating a seiche. By 28 Dec, sea level in the Gulf of Bothnia dropped, and the resulting seiche oscillation in the Gulf of Finland created a strong sea-level gradient across the AS, one that was able to drive currents against the now-weakened local wind." This separates the two basins' contributions and uses correct timing.

- **(line 354)**: *"compared to the SW corned (Föglö)."* – **"corned"** is a typo; should be **"corner."**

- **(line 355–358)**: *"opposing the sea level gradient to the wind direction only weakens the effect of local winds, whereas currents still follow the direction of the winds. During storm Toini on 11 January 2017 (Figure A2), the winds… The maximum currents measures were 99 cm s$^{-1}$, being*

*the third strongest event and had the longest lasting duration of currents over 40 cm s⁻¹ (35 h).”*
Consider rephrasing and grammar improvements.  Instead of "*currents measures"*  "current was".

- **(line 364)**: *“the currents did not begin to follow the sea level gradients before the wind speeds dropped below 10 m s⁻¹.”* – Perhaps use **“until”** instead of **“before”** here.
- **(line 370)**: Once again referring to Fig. 12 and pointing to  'dark green'  wind speeds relation to current speed is hard to follow from figure.
- **(line 379)**: *“Continuous atmospheric pressure was only available from 2007 onwards (Sect. 2.2).”* – Minor phrasing issue: add **"data were"** after “pressure” for clarity.
- **(line 382)**: *“Three-hourly measurements from 1991 to 2020 show that the mean wind speed is 7.5 m s⁻¹ in Utö and 6.9 m s⁻¹ in Isokari, and 99%, respectively, being 18 m s⁻¹ and 16 m s⁻¹.”* – The part after the comma is confusing. It appears to refer to the 99th percentile wind speeds, which could be used for clarity in further text aswell (eg **and the 99th percentile wind…** ).
- **(line 404)**: *“the event on 29 December 2016 ”* Correct the date if needed.
- **(line 417)**: *“about 7 times the distance than between Utö and Isokari.”* – Should be **“seven times the distance between Utö and Isokari.”** (drop “than”).
- **(line 485)**: *“...shows that neither of these forcings grows strong enough….”*
Maybe change **“forcings grows”** to **“forcing becomes”** or **“none of these forcings is strong enough to induce currents as large as wind can.”** Minor grammar.
- **(line 487)**: *“23 m s⁻¹ winds could be estimated to include currents well above 50 cm s⁻¹.”* – perhaps use **"induce currents"** or **"generate currents"** instead of **"include"**.
- **(line 488)**: *“As we for the first time have such a large data set of measured currents from the AS, we were able to catch extreme current speeds…”* –  The sentence has awkward word order and mismatched verb tenses, making it unclear and grammatically incorrect. Consider rewriting the sentence.
- **(line 490)**: *“Rantanen et al. (2024) has shown”* – **“has”** should be **“have”** (plural verb for “et al.”).
- **(line 498)**: *“Measurements in the AS can be divided into two types based on current magnitudes: more open sea areas with a mean surface layer magnitude of around 8 cm s⁻¹ and narrow long straits with mean surface magnitudes of around 14 cm s⁻¹.”* – Clear summary, but perhaps adding **"surface layer current"** would increase readability.
- **(line 502)**: *“ the measurement values were below 10 to 20 cm and 99% of the values were around half of the maxima.”* – Add **“per second”** after “10 to 20 cm” for units consistency (assuming “cm” alone is a typo and should be cm/s).
- **(line 505)**: *“whereas in the southern most Utö stations....”*
**“southern most”** should be **“southernmost”** (one word).
- *Figure A1 & A2 captions:* Both start with *“Time series”* descriptions. In Figure A2’s caption it reads *“Time series the third strongest event…”* –  add “of the”. Similarly ensure Fig. 13 and 14 captions say “Time series **of the**…event” for grammatical completeness. These are minor but noticeable errors in figure captions.
- **(line 528)**: *“provided expertise and incite to the conclusions from their area of expertise.”* – perhaps author meant "insight" instead of  **“incite”**  :)

---

## Author Comment (AC1)

**Author's reply to referee comments**

Kanarik, H., Tuomi, L., Alenius, P., Miettunen, E., Johansson, M., Roine, T., Westerlund, A., and Kahma, K. K.: Observed currents in the Archipelago Sea, EGUsphere [preprint], https://doi.org/10.5194/egusphere-2025-1101, 2025.

We thank the referee for taking the time to read and comment our manuscript thus helping us to improve our work. Below, referee comments are displayed with italic font. Our author replies are with plain blue text.

**RC1, Anonymous Referee #1, 30 Apr 2025**

*This is a manuscript describing current measurements in the Archipelago Sea in the northern Baltic Sea. Observations time series from various times and locations are analyzed and related to forcing from winds, sea levels and air pressure, and it is found that winds are the main forcing component, but that sea levels can weaken or strengthen currents under certain conditions, mainly when the sea level in the Bothnian Bay is low and that in the Gulf of Finland is high due to seiching motions. Generally it is a well written and well performed study that deserves publication.*

*One issue is that it makes little sense to compare the maximum value between different observation series since it is a function of record length as well as ensemble length. It would be better to present a percentile value, e.g. 99% or 99.9% or something like that, e.g. of hourly mean current speeds, in order to make the numbers more comparable.*

Thank you for this suggestion, as well as for the positive overall assessment. We have updated the values in Chapter 3.5 "Seasonal statistics of the current magnitudes" and removed the comparison of maximums from the very beginning. We also updated Figure 10 to include hourly mean magnitudes and 99% instead of maximum values. To enhance the persistency analysis, as suggested by second reviewer, we also added information of completeness of each season as well as seasonal persistence values from each station. In Chapters 3.1 to 3.4 where we discussed the characteristics of measured currents based on the area of the measurement, we think it is appropriate to keep the original measured values, especially as we focus to describe the measurements and the general directionality of the currents, driven by the geometry of the area.

*Detailed comments:*

We thank the author for noticing these mistakes in the text. In case of a larger correction, we have included the original and revised sentences to this answer.

- *Line 14: basing -> basins, current -> currents*
  Corrected

- *Line 15: dirction -> directions, related -> relative*
  Corrected
- *Line 66: published -> unpublished ?*
  Corrected
- *Line 90-91: move "bottom mounted" to before "Teledyne" since Teledyne do probably not sell bottom mounted instruments* 😊
  Corrected

- *Line 103: Unclear what this means. Was the instrument compass calibrated for magnetic variation or how was it taken into account?*

  Sentence "Magnetic variation was taken into account during the deployment setup." was changed to "Magnetic variation was corrected by calibrating compasses prior to deployments and applying up to date magnetic declination correction."

- *Lines 142- 143: Unclear what is meant here. Winds are weakened between NNE and S but E (which is within this interval) is overestimated??*

  We re-checked this information from the weather station, and it was written in unclear way. By "*between NNE and S*" we meant area from west between S and NNE. We changed the sentence "*The size and shape of the Isokari island weaken the winds from NNE to S directions; however, winds from E are slightly overestimated due to the height of the measurement device.*" to:

  "**As wind measurements are made on the eastern edge of the Isokari island,** the size and shape of the island weaken the winds from **the western sector between the S and NNE directions**; however, winds from E are slightly overestimated due to the height of the measurement device."

- *Lines 157  - 159: Unclear text. Please clarify.*

  Sentence "*As in this study, we are interested in the sea level tilt driving the currents, we used hourly sea levels relative to the theoretical mean sea level for statistical analysis over climatological standard normal period of 1991 to 2020 and half-hourly measurements for analysis on simultaneous current and sea level measurements.*" was changed to the following format to avoid repetition with text in an earlier chapter and make it more clear. The first sentence of the chapter "To evaluate the relation between currents and the sea level differences over the area, we analysed hourly instantaneous sea level data from tide gauges at Rauma ..." was also corrected :

*"To evaluate the relation between currents and the sea level differences over the area, we analysed **instantaneous sea level data** from tide gauges at Rauma …* **The sea level values used are given relative to the theoretical mean sea level. For statistical analysis of typical sea level difference, we used hourly instantaneous data over climatological standard normal period of 1991 to 2020.***"*

- *Line 190: "Halocline" -> "The halocline". Please also check similar issues other places in the text.*
  Corrected

- *Lines 236-240: This is interesting, since it must mean that water is added along the length of the channel. Any speculations about where this water is taken from?*

  This refers to lines: *"The narrowness and length of the strait strengthen currents along the flow direction, so that southward travelling currents are strongest at the southern edge of the strait and vice versa, as seen from simultaneous measurements of currents in Norrgrundet and Gloskär (Fig. 8). The distance between the stations is around 28 km (15 nautical miles) and within this distance, the current magnitude increases to around double in the northern end of the channel with northward currents. For southward currents, the increase in magnitude along the channel is slightly less. "*

  We see that our sentence here was bit misleading here. We wanted to emphasize that in these straits there are two factors that increase the flow strength: first the narrowness of the strait compared to the more open areas, where the waters push into the strait strengthening the flow speed,  and second the length of the straight enables growth of flow speed also by winds. We have updated this part as follows:

  **"The flow speed increases as the water flows into the narrow strait from the more open areas of the north and south. Additionally, wind can further increase the flow speed along the channel.  As a result,** southward travelling currents tend to be strongest at the southern edge of the strait and vice versa, as seen from simultaneous measurements of currents in Norrgrundet and Gloskär (Fig. 8). …"

- *Line 267: This -> these*
  Corrected

---

## Author Comment (AC2)

**Author's reply to referee comments**

Kanarik, H., Tuomi, L., Alenius, P., Miettunen, E., Johansson, M., Roine, T., Westerlund, A., and Kahma, K. K.: Observed currents in the Archipelago Sea, EGUsphere [preprint], https://doi.org/10.5194/egusphere-2025-1101, 2025.

We thank the referee for taking the time to read and comment our manuscript thus helping us to improve our work. We see that your contribution improved the readability and thus the reachability of our work. Below, referee comments are displayed with italic font and line numbers refer to revised version of the manuscript. Our author replies are with plain blue text.

**RC2, Anonymous Referee #2, 02 May 2025**

**Review of "Observed currents in the Archipelago Sea"**

*This manuscript presents a comprehensive observational study of currents in the Archipelago Sea (AS), an extremely complex coastal area in the Baltic Sea with myriad islands and narrow straits. The authors have compiled Acoustic Doppler Current Profiler (ADCP) measurements from 10 locations over roughly two decades of short campaigns - a valuable dataset that fills a longstanding gap in Baltic Sea observations.*

*The study is highly relevant to Ocean Science, addressing important questions about coastal circulation patterns and extreme currents in an archipelago environment. The manuscript's overall scientific quality is good: the ADCP measurements are well described with appropriate quality control, and the results are presented thoroughly with substantial interpretation. The findings are significant for both scientific understanding and practical applications, as the authors quantify typical current magnitudes and demonstrate the conditions for powerful currents. Notably, the paper identifies local wind as the primary driver of currents but also highlights how basin-scale seiches and local sea-level differences can generate strong flows when local winds are weak or unfavorably oriented. However, the authors' claims would be more convincing if they were better supported by the available observational data or complemented by modeling, which could help clarify the intervariability among parameters. Overall, the manuscript is well structured and written in generally clear language, although the clarity of the writing and the presentation of the figures could be further improved. It represents a useful and original contribution that fits well within the scope of Ocean Science.*

We thank the reviewer for constructive comments that have helped us to improve our manuscript, as well as for the positive assessment. We have revised the manuscript to improve the clarity of the writing, enhance the presentation of the figures, and refine the interpretation of the results based on your comments. We believe these improvements have made the manuscript clearer and more compelling.

The current dataset used in this manuscript is the most comprehensive one from this area and we have used meteorological and other marine observations (e.g. temperature and salinity profiles and tide gauge data to support our analysis. Although, modelling results for this exist they only cover a short few years' period. Therefore, we decided to exclude them from our analysis. We agree that a more comprehensive modelling study for this area would be needed covering longer period would be needed for more comprehensive analysis, but it is out of scope of this study.

***General comments***

***Data Gaps and Seasonal Bias****: Some deployments suffered from significant data gaps. For instance, at Fölskär only 35% (or to be corrected?) of near-surface data was obtained due to summertime acoustic issues (implied around lines 117–126). The authors should discuss whether such gaps might bias the analysis of seasonal currents. Do missing summer measurements (e.g., June–July at Fölskär) affect the calculated mean speeds or directional distributions for that location? A brief comment on if and how these data gaps were handled would be useful.*

We replaced the quick notion on the data quality at the end of Section 3.1:
*"Note that a large part of the simultaneous surface measurements were missing in Fölskär, making this layer less comparable to other layers. The mean current speed in this intermediate layer was 7 cm/s at both stations."* with bit more elaborated version to highlight this:

*"It should be noted that the surface layer measurements were only available from the later Fölskär deployment (Table 2, ID SVM2S) from April to August and even these had notable data gaps due to lack of scatterers in the surface layer during the day. The short measurement period together with less available data at the layer above the seasonal thermocline largely explains the overall smaller current magnitudes at the Fölskär site as elaborated more in Section 3.5."* We also elaborated on the missing data at the Föskär site as requested in the later detailed comment for line 131 in Section 2.1.1 "Quality Issues". We also added a notion on how the missing data was handled in the analysis to highlight this to the readers [Section 2.1, line 96-98]:

"Bad quality data was marked as nan in the analysis, thus leaving data from unmeasured areas completely out of the analysis. Data sets that have a large amount of missing data are marked in the figures to highlight the higher uncertainty of the given values."

Based on these comments, we made changes to Section "3.5 Seasonal statistics of the current magnitudes" and Figure 10 (**see below**). As noted by the other reviewer these values would have been more comparable if we used hourly data for seasonal analysis. Thus, we updated the analysis using hourly means of current magnitudes and added information on the data completeness for each season to improve the interpretation of

the given values. As the maximum values were already stated earlier in the site descriptions, we replaced these values with 99% to better showcase the general features of the current magnitude distribution.

We also added a reminder on the data gaps to this section [line 281-284]:
"Note also that not all stations have measurements that range through all seasons. For example, very few measurements are available for Lövskär during spring (Fig. 10b), Norra Bredan during summer (Fig. 10c) and Fölskär during autumn (Fig. 10d). Moreover, several stations do not have measurements for all four seasons, which limits the temporal comparability of the results (Fig. 10)."

[Figure]

**Figure 10.** Seasonal statistics calculated from hourly mean current magnitudes over all depth bins. The value 90% and 99% represent the 90th and 99th percentile of the current magnitudes. Persistency values are given seasonally to each station's surfacemost (Pbm, bolded values) and bottommost (Pbm) depth cells (corresponding depths can be found in from Figures 5, 6, 7 and 9). Additional persistency value is given for the stations at the southern edge of the AS (Fölskär and Utö) with the occasional halocline given between Psm and Pbm -values. Completeness of the dataset of each season is given in top x-axis as a blue bars, where value 1 stands for at least one complete season of measurements, and for example, 0.5 indicates that during this season the measurements only cover around 1.5 months out of the three per in one season.

*Persistency Metric Usage: The persistency (P) metric is defined in Section 2.3 (line 165–170) as a measure of how consistently currents maintain one direction. However, P is mentioned only briefly in the text, mostly through unnecessary numerical statements without further explanation or discussion. It would strengthen the manuscript to report*

*and interpret P values for the different regions. For example, how persistent are the flows in narrow straits versus open areas or southern vs northern AS? Reasons for low P (high or low frequency oscillations) including a short commentary on persistence (perhaps in Section 3 or 5) would give a quantitative sense of directional variability and complement the qualitative descriptions.*

We felt that more detailed discussion on the persistency values required also information on the seasonal values, so we calculated and added this information to Figure 10 for each season and each presented depth in earlier current rose figures (Pbm/Psm [%]), where we added to lines 325-327:
"Like in Norrgrundet and Gloskär, most stations showed slightly higher persistency values near the bottom compared to the surface layer (Fig. 10). When examining seasonal persistency, the measurements show substantial variation not only between different seasons but also between nearby stations that were measured during different years."

Based on these values we added a short chapter on persistency of the currents at the end of conclusions showing the motivation of that concept, but further analysis of current persistency is left out of the present paper. [line 467-474]:

"An early study on general circulation of the Baltic Sea that was based on current observations from lightships (Witting 1912) indicated that there are seasonal differences in the persistency of surface layer currents in the coastal regions of the Bothnian Sea and in the Gulf of Finland. That study also concluded that there is outflow from the Bothnian Sea through the Archipelago Sea in early summer and inflow to the Bothnian Sea through the Archipelago Sea in autumn. We calculated the seasonal persistency of currents in the Archipelago Sea (results shown in Figure 10). Our results do not confirm such a clear resultant circulation pattern to be present in the AS. Especially in autumn the persistence of the currents seems to be smaller. This is in accordance with the rather low persistency of winds in monthly statistics of 2000 – 2016."

***Seiche- vs Wind-Driven Flows:*** *A major finding is that about 75% of strong current events align with local wind, while ~25% oppose it due to sea-level gradients (lines 319–322). This is an important result for understanding forcing mechanisms, however this is hard to see from the later analysis of Fig 12-14, therefore all of the conclusions and statements on these figures are a bit vague. Further the authors might consider expanding the discussion on how to predict or identify these sea-level-driven events as the navigation safety is mentioned multiple times. For instance, could a combination of low local wind ($<10 \text{ m s}^{-1}$) and a large Gulf of Finland seiche be a practical indicator of an upcoming current reversal?*

We added an explanation of these events to Line 336-339: "These events were defined by studying the simultaneous winds during high-current events and classifying the events based on the prevailing wind direction. As this dataset included numerous high current events, we chose to present a few time specific events that demonstrate the main features of the different cases identified."

In the discussion, we also elaborated the usage for navigation in the following way (from Line 524-529):

"When considering safe navigation in the Archipelago Sea, these measurements show that high currents typically occur with high and storm wind events. But considering also the events where high current speeds occur also during relatively calm weather induced by sea level differences it is difficult to make any simplified cause-and-effect relationship solely based on wind forcing, as also shown in Fig 12. For issuing warnings of high current event a high-resolution operational model would be needed."

*Contextualizing Findings: The discussion could better contextualize these results relative to previous knowledge and broader implications. The authors do compare their measurements to past campaigns and models (lines 443–451), noting similar behavior in other straits and agreement with model simulations. It would be worthwhile to explicitly highlight what is new. For instance, is this the first direct evidence of >100 cm s$^{-1}$ currents in the Archipelago Sea (confirming model predictions)? Do the results alter our understanding of AS circulation, or mainly reinforce existing model expectations? A few sentences synthesizing how this extensive dataset advances knowledge would be valuable. Additionally, since safety of navigation is a motivation (lines 225-226), the authors might comment on how their findings could be used by mariners or somehow integrated into warning systems in the future.*

We have broadened our discussion on these issues in the following places in discussion.

Line 462: "Our measurements for the first time show measured currents of 1 ms−1 in the AS."

Lines 474-481: (new text is bolded): "In general, our measurements show a very similar directional distribution of the currents as presented in the model simulations by Miettunen et al. (2024), although the measurements and model results cover different time periods. **The focus of these model studies is in the overall circulation and transport dynamics of this area and are not as such comparable for our findings based on measured data.** Combining information with measurements with suitable circulation modelling will in the future lead to a more comprehensive overview of the circulation dynamics also in those areas from which we lack measurements. **Our findings encourage to take a closer look at single storms and large oscillation**

**events in the surrounding basins (like in Fig. 13) to quantify their significance to the water exchange through the AS."**

To further elaborate the persistency value mentioned in the text we added following notion to discussion:

***Title could be refined:*** *it is accurate, but overly broad and generic. It doesn't reflect the depth of the analysis or the study's key contributions. A more descriptive and specific title would help the paper stand out and communicate its scientific focus better.*

Thank you for this suggestion. We decided to change the title to: **Currents and their Drivers in the Archipelago Sea: Insights from ADCP measurements**

***General comment regarding figures:***

- *Consider adding letters to figures instead of (lower right panel). See the "Figure content guidelines:" in journals' submission guidelines.*

  As suggested, figures were updated to follow this guideline

- *Avoid crowded scatter plots, where the majority (?) data is overplot.*

Thank you for constructive comment about Fig. 12. We have redrawn the Figure so that it better shows the populations of interest yet still containing the whole dataset. We hope that this new way of representation makes the Figure easier to read. We also removed the panel showing wind directions as the information was not necessary to show in a figure, thus leaving more room for the more interesting information. New figure shown below:

[Figure]

**Figure 12.** Norrgrundet current magnitudes against sea surface height difference between Hanko and Rauma. The data is divided by the main current directions: northward currents (a) and southward currents (b). The corresponding wind speed is presented as a colour and main wind direction is presented by marker arrow direction. Positive sea level difference drives currents northward and vice versa. Northward direction is defined as > 315∘and ≤ 45∘and the other three main compass directions are defined correspondingly. N indicates the number of different data points in each scatter. The data is sorted so that events during the smallest wind speeds are shown on the top. Northwards currents drive by a large sea level difference caused by the oscillation on the surrounding basins are highlighted with an ellipse in panel (a).

***Specific Comments:***

- *(line 5): "lack of quality ensured measurements" – This phrasing is odd. It should likely be "lack of quality-assured measurements", since the authors mean there were no previously quality-controlled current observations.*
  Corrected

- *(line 13): "northern end of the long NE strait" – The term "NE strait" is used without prior definition. Readers might not know NE means north-eastern here. Consider spelling out or referring to it as "the north-eastern strait" for clarity.*
  Corrected to use the word north-eastern instead of NE

- *(line 14): "surrounding basing" – Typo, "basing" should be "basins." (e.g., "oscillations in the surrounding basins").*
  Corrected

- *(line 15): "non optimal direction related to the straits" – Awkward phrasing. It would read better to use a hyphen in "non-optimal" and clarify "related to" as "relative to" or "with respect to."*

Corrected to "… from non-optimal directions with respect to the straits."

- *(line 30): "using a large network of light ships" – Consider clarifying "light ships" if this is a historical term (lightships). For modern readers, a brief note that these were essentially floating lighthouses used for measurements might help, but this is minor.*

  Added: "The first current measurements in the Baltic Sea were conducted more than a century ago using a large network of lightships, **which were essentially floating lighthouses that were also utilised for marine observations** (refs). "

- *(line 66): "have been largely unpresented and published before this paper." – This seems contradictory. Likely a mistake: perhaps "unpresented and unpublished" was intended.*

  Yes this was a mistake, corrected

- *(line 69): "inner parts of the AS and concluded that the interactions are extremely complicated due to the heterogeneity of the area." – Suggest adding a explanation if possible. It's understandable, but "extremely complicated" interactions could be elaborated in manuscript, since the reference is a report in non-English and very old one. Perhaps (if available) refer to some recent study which elaborates the "extreme complexity.*

Unfortunately, this is the only study on this topic in the inner archipelago area. We however elaborated this sentence slightly in the following way

"Virtaustutkimuksen neuvottelukunta (1979) evaluated the effect of winds on currents in the inner parts of the AS and concluded that the interactions are **very** complicated due to the heterogeneity of the area. **They found that** the connection between wind direction and current direction was clearest when the winds **were aligned along the channels and blowing** from the same direction for longer periods."

- *(line 76): "even 15 m/s SW winds were not able to induce strong currents in this area" – Perhaps specify "in that area" (Lövskär cross-section) to make it clear you are referring to the specific site studied by Kanarik et al. (2018), not the whole archipelago.*

Sentence "This was also demonstrated by the analysis of current measurements in the Lövskär cross section, where currents were shown to be very sensitive to even small changes in wind directions and that even 15 ms−1 SW winds were not able to induce strong currents in this area (Kanarik et al., 2018)." was rephrased to "… were not able to induce strong currents in **that cross section** (Kanarik et al., 2018).

- *(Table 1): The table of ADCP moorings has some formatting issues.*
  - *For example, the second entry "SVM2S 25 Apr – 25 Aug 2006" lacks a location name. Is this the Fölskär measurement (assuming from Fig 2.)?*

*Please ensure each dataset is clearly named (or clearly grouped) in the table for cross-reference.*

o *Also, the note 'device changed on XXX' should be included as a footnote or placed in an additional column, since the current column is intended specifically for the observation period.*

Thank you for pointing this out as we did not realise this can be unclear for readers. Stations in table that had repeating measurements were named as "Fölskär" and in repeting lines "- Fölskär" to keep table easier to read. Notion of device change is moved to footnote. We also edited Table 2 caption to so that notion with a * was moved to be footnote below the table.

- *(line 125): Devise should perhaps be device.*
  Corrected

- *(line 127): without the technical understanding or description about the technology behind ADCP devices it is hard to understand what are the 'ensambles' and 'solutions' that ADCP is unable to calculate.*

The sentence "The depth of the Fölskär site (115 m) was too deep for the 300 kHz ADCP as the devise was mostly unable to acquire125 measurements in the upper 50–60 m of the water column. In the first data set (measured in 2004), the maximum number of measurements was missed at depths of 36–38 m, where ADCP was unable to **compute the velocity solution** from 68% of the **ensembles**." was edited to: "…, where ADCP was unable to **compute the current velocities** from 68% of the **measured times**."

- *(line 131): The description of data return at Fölskär is a bit hard to parse. It says only 35% of near-surface measurements were successful above the thermocline. Though I can not conclude from Fig. 2, where the amount of "good data" seems to be significantly higher.*

When looking back, this part should have had more clarifications. These lower numbers of good data occurred at certain depths (36—38 m) and as the site was rather deep (115) and most of the measurement at the lower layers were successful, the overall amount of good data (presented in Fig. 2) is higher. We added notion to the end of the paragraph:
"**However, there were almost no missing data in either of the measurements below the 60 m ranging to almost 110 m depth.**"

- *(line 133): "vertical velocities and thus error velocities were exceptionally high at the bottommost 4 bins." – Good that this was caught. The fix (removing bottom four bins) is fine, but maybe mention in the Table 2 or text (perhaps ref ID) which dataset instead of "Norrgrundet second deployment" this concerns, for clarity.*

This was briefly mentioned in the Table 2 "Quality issues" –column as the "bottom", but we agree that it would be good to highlight further. We now added reference to the measurement ID in the beginning of the sentence "In the second Norrgrundet dataset **(ID S2-17, Table 1)**, …"

- *(line 150): "Continuos simultaneous data of atmospheric pressure from multiple ASW stations…"*
  - *– Two issues here. "Continuos" is a typo; it should be "Continuous." Also, "ASW stations" appears to be a typo for "AWS stations" (Automatic Weather Stations). Please correct these to avoid confusion.*

  Corrected

- *(line 163): "This data is available…" – Technically, "data" is plural. Perhaps say "These data are available…" (also at line 147). Consistent plural use would be preferred (e.g., "Data were collected…" rather than "data was").*

  Corrected

- *(line 168): The formula for persistency (P) is presented. Ensure that all symbols ($\sum$, n, N, u_n, v_n) are properly defined in the text. Currently, "where u and v are the eastward and northward velocity components…" is given. Perhaps also clarify that n indexes each observation in the time series and N is the total number of observations, for completeness.*

  We added the following information to the Section 2.3:

  "… where un and vn are the eastward and northward components, of the current velocity at time step n. N stands for the total number of observations in the time series, n indexes each observation from 1 to N, and $\sum$ denotes the sum over all time steps from n = 1 to N ."

- *(line 172–176): The definition of four geographic sub-regions is useful. However, it might help to explicitly list which measurement sites belong to each region (perhaps in the caption of Figure 1 or in Section 3.1–3.4 headings or in Tables). Currently the reader must infer from Section 3.1 text which sites are "southern edge." A direct mapping (e g on Fig) would improve clarity.*

This was a great idea! We decided to add an "Area" column to Table 1 as well as colour coding the stations based on the area to Figure 1, showing the measurement locations.

- *(line 185): "3.1 Currents in the southern edge of the archipelago" – Consider capitalizing Archipelago Sea when referring to the region (consistency issue). Also "southern edge" vs "southern part" – just ensure consistency with earlier nameings.*

  Titles were changed from original format "Currents in the southern edge of the **archipelago**" to suggested "Currents in the southern edge of the **Archipelago Sea**"

- *(line 186): "the occasional halocline and seasonal thermocline divide the water column" – verbagreement: "divide" should be "divides". Or rephrase the sentence for clarity.*
  Corrected
- *(line 206): "3.2 Currents inside the archipelago" – Same naming issue as earlier 3.1 paragraph. Perhaps use "central archipelago" to align with the earlier description of Area 2.*
- Corrected to "3.2 Currents in **the central Archipelago Sea**"
- *(line 220) "higher persistency at Söderkobb compared to the other two stations are explained by the seasonality of the measurements (Sect. 3.5)." raises the question of whether this higher persistence is actually due to missing observations during colder periods at Söderkobb, unlike the other stations. It would be helpful to elaborate on this point in Section 3.5 and discuss whether the observed persistence is a statistical artifact resulting from data gaps or a true physical feature. In a similar manner, it would be worthwhile to reassess the rest of the analysis to ensure that patterns attributed to physical processes are not instead influenced by data gaps or seasonal biases.*
  The following questions and comments have already been answered in the general comments section
- *(line 313): "reached magnitude up 30 cm s $^{-1}$ at Norrgrundet" – Missing a word: "up to 30 cm/s."*
  Corrected
- *(line 319): "There were all together 224 events during c. 30 months…" – "all together" should be "altogether" (one word) in this context. Also consider replacing "c. 30" with "approximately 30" for clarity (casual readers may not recognize "c." as "circa").*
  Corrected
- *(line 327-333): It seems to be one of the key results. However it is hard to interpret from the Figure 12. As much as I can read from the figure it seems that the strongest northward currents (>100 cm/s) are mainly related to "winds to west" rather than "to north". Same for southward currents where "winds to east" could be related to strong currents. Perhaps some multi-linear regression or clustering technique could be used to make more solid estimates with interrelationships between current speeds, sealevel and wind characteristics. Currently these inter-relationships are a bit vague and almost impossible/controversial to see from the colorful scatter plots.*

When presenting the first timeseries (Fig 13) we now added an explanation of these easterly and westerly winds in the scatterplot [Line 354-356]:

"The current speed grew until the winds turned to a less optimal direction related to the strait (Fig. 13). The growth of high current events was most often stopped by the turning of the wind, thus the maximum values presented in Fig. 12 often measured simultaneous winds from either east or west."

- *Figure 8. Add Isokari AWS, if it fits to the region*
  We slighly increased the area to fit Isokari AWS to the figure
- *Figure 12. It is hard to interpret the relationships given in text from the figure. Consider better phrasing, additional annotation - or completely redoing the figure. Too much information makes the reading hard - and even controversial (see previous comment regarding lines 327-333).*
  - *Consider focusing only on the key data that effectively delivers the message, rather than including all measurements, which can overwhelm the reader. If, during revision, you find that all data points are indeed necessary, avoid plotting them all on top of each other, as this currently results in displaying only a partial segment at the end of the time series. Additionally, it would be clearer to use a consistent approach to wind direction within a single figure - either 'winds to' or 'winds from' - to avoid confusion, or justify the use of different notation.*

We edited the figure by deleting the left most panel form the figure (with wind directions as colour) and changing the colours to make the different wind speeds stand out better. We also sorted the data so that the values during the smallest and largest (over 16 m/s) winds speeds are on top. We tested multiple different options with this figure by dividing it to different subplots based on the wind magnitudes and directions, removing some of the data etc. but the chaotic interactions of these forcings would most likely require a completely different approach as you commented.

We however think that this edited figure now shows the message we want to give in this paper: 1) There is no clear linear interaction with wind now sea level in the complex AS area. Neither high wind nor sea level difference can alone be taken as indicator of high currents in the area as they are very interconnected and (as shown from time series examples) the highest currents form when the ratio of these two is most suitable. 2) There are clear set of events driven by the sea level differences between the surrounding basins of the AS, that can grow strong event with opposing winds. We now highlighted this area with an ellipse for readers and instead of "light colours" we now reference this area.

- *(line 334): "During the strongest current event, with a magnitude of 115 cm s$^{-1}$ on 22 February 2017 (Fig. 13), both the wind and the sea level forcing drove the northward flow..." – The phrase "sea level forcing" might be unclear. Maybe say "sea level gradient" or "sea-level setup" to be explicit.*
  Changed to sea level gradient

- *(line 336): "In addition to external forcing, the narrowness of the channel further amplified the flow (Fig. 8)." – Here "external forcing" refers to wind and pressure presumably. This is fine, but perhaps specify: "In addition to wind and sea-level pressure forcing..." for clarity, since "external" could be interpreted as outside the strait.*
  Corrected

- *(line 337): "SSE winds with around 20 m/s speed blew for several hours and the sea level was 9 cm higher in Föglö than in Rauma and Hanko" – It would be clearer as: "SSE winds of ~20 m/s blew for several hours, and during this time the sea level at Föglö was about 9 cm higher than at Rauma and Hanko...". This explicitly states which location had the higher water level.*
  Corrected

- *(line 339): "The second strongest event, with a magnitude of 104 cm s$^{-1}$ on 28 December 2016 (Fig. 14), followed the sea level gradient over the AS and was formed while the wind was opposing the strong sea level gradient." – There is potential confusion with dates (see next comment). Assuming 28 Dec 2016 is correct, the wording "followed the sea level gradient" could be rephrased as "occurred in response to a sea-level gradient across the AS" or "was driven by a sea-level difference across the AS, with wind blowing in the opposite direction."*
  Thank you, this is much better wording. We used the latter suggestion.

- *(Fig. 14 caption): There is a discrepancy in dates for the second strongest event. The text says 28 December 2016, but the Figure 14 caption caption refers to "(29 Jan 2016)" – which seems incorrect. This is confusing and likely an error. Please clarify the timeline of events:*
  Corrected, it should have indeed read 28 December.

- *(line 342): "wind speed had degreased below 10 m s$^{-1}$." – "degreased" is a typo; should be "decreased."*
  Corrected

- *(line 344-346): "winds packed water to the end of the gulf on 24 December 2017 (seiche). Once sea level simultaneously fell at the Gulf of Bothnia on 28 Dec, the simultaneous seiche at the Gulf of Finland created such a strong gradient over the AS that it was able to oppose the weakened local wind." – The dates here are likely wrong given context. Assuming it should be late 2016. Consider rephrasing eg. "winds had piled up water at the head of the Gulf of Finland on 24 Dec 2016, initiating a seiche. By 28 Dec, sea level in the Gulf of Bothnia dropped, and the resulting seiche oscillation in the Gulf of Finland created a strong sea-level gradient across the AS, one that was able to drive currents against the now-weakened local wind." This separates the two basins' contributions and uses correct timing.*

Thank you. The year was again written wrong in text. We used the suggested wording as it is much nicer and clearer for the reader.

- *(line 354): "compared to the SW corned (Föglö)." – "corned" is a typo; should be "corner."*
  Corrected

- *(line 355–358): "opposing the sea level gradient to the wind direction only weakens the effect of local winds, whereas currents still follow the direction of the winds. During storm Toini on 11 January 2017 (Figure A2), the winds... The maximum currents measures were 99 cm s$^{-1}$, being the third strongest event and had the longest lasting duration of currents over 40 cm s$^{-1}$ (35 h)." Consider rephrasing and grammar improvements. Instead of "currents measures" "current was".*
  Changed to During storm Toini on 11 January 2017 (Figure A2), the winds in Isokari **were** up to 22.5 m s$^{-1}$, and stayed around 20 m s$^{-1}$ for more than 18 hours. The maximum current **was** 99 cm s$^{-1}$...

- *(line 364): "the currents did not begin to follow the sea level gradients before the wind speeds dropped below 10 m s$^{-1}$." – Perhaps use "until" instead of "before" here.*
  Corrected

- *(line 370): Once again referring to Fig. 12 and pointing to 'dark green' wind speeds relation to current speed is hard to follow from figure.*
  We changed the colormap to hopefully better

- *(line 379): "Continuous atmospheric pressure was only available from 2007 onwards (Sect. 2.2)." – Minor phrasing issue: add "data were" after "pressure" for clarity.*
  Corrected

- *(line 382): "Three-hourly measurements from 1991 to 2020 show that the mean wind speed is 7.5 m s$^{-1}$ in Utö and 6.9 m s$^{-1}$ in Isokari, and 99%, respectively, being 18 m s$^{-1}$ and 16 m s$^{-1}$." – The part after the comma is confusing. It appears to refer to the 99th percentile wind speeds, which could be used for clarity in further text aswell (eg and the 99th percentile wind... ).*
  Changed to "Three-hourly measurements from 1991 to 2020 show that the mean wind speed is 7.5 ms−1 in Utö and 6.9 ms−1 in Isokari, **and respectively 99th percentile of the wind speeds** being 18 ms−1 and 16 ms−1."

- *(line 404): "the event on 29 December 2016 " Correct the date if needed.*
  Corrected to 28 December 2016. Thank you.

- *(line 417): "about 7 times the distance than between Utö and Isokari." – Should be "seven times the distance between Utö and Isokari." (drop "than").*
  Corrected to: "... which have about 7 times the longer distance than between Utö and Isokari."

- *(line 485): "…shows that neither of these forcings grows strong enough…." Maybe change "forcings grows" to "forcing becomes" or "none of these forcings is strong enough to induce currents as large as wind can." Minor grammar.*
  Corrected to "Comparison to idealised current speeds over the area also shows that none of these forcings is strong enough to induce currents as large as the wind can."
- *(line 487): "23 m s $^{-1}$ winds could be estimated to include currents well above 50 cm s $^{-1}$." – perhaps use "induce currents" or "generate currents" instead of "include".*
  Corrected to "generate currents".
- *(line 488): "As we for the first time have such a large data set of measured currents from the AS, we were able to catch extreme current speeds…" – The sentence has awkward word order and mismatched verb tenses, making it unclear and grammatically incorrect. Consider rewriting the sentence.*

Changed to: "This longer data set of measured currents allowed us to catch extreme current speeds caused by sea level fluctuations in the basins surrounding the AS."

- *(line 490): "Rantanen et al. (2024) has shown" – "has" should be "have" (plural verb for "et al.").*
  Corrected
- *(line 498): "Measurements in the AS can be divided into two types based on current magnitudes: more open sea areas with a mean surface layer magnitude of around 8 cm s $^{-1}$ and narrow long straits with mean surface magnitudes of around 14 cm s $^{-1}$." – Clear summary, but perhaps adding "surface layer current" would increase readability.*
  Added as suggested
- *(line 502): " the measurement values were below 10 to 20 cm and 99% of the values were around half of the maxima." – Add "per second" after "10 to 20 cm" for units consistency (assuming "cm" alone is a typo and should be cm/s).*
  Corrected
- *(line 505): "whereas in the southern most Utö stations…." "southern most" should be "southernmost" (one word).*
  Corrected
- *Figure A1 & A2 captions: Both start with "Time series" descriptions. In Figure A2's caption it reads "Time series the third strongest event…" – add "of the". Similarly ensure Fig. 13 and 14 captions say "Time series of the…event" for grammatical completeness. These are minor but noticeable errors in figure captions.*
  Corrected

- *(line 528): "provided expertise and incite to the conclusions from their area of expertise." – perhaps author meant "insight" instead of "incite" :)*
  Corrected :)

---

## Author Response (AR2)

**Author's response to the editor**

Kanarik, H., Tuomi, L., Alenius, P., Miettunen, E., Johansson, M., Roine, T., Westerlund, A., and Kahma, K. K.: Currents and their Drivers in the Archipelago Sea: Insights from ADCP measurements, EGUsphere [preprint], https://doi.org/10.5194/egusphere-2025-1101, 2025.

**Public justification (visible to the public if the article is accepted and published)**:

 Dear Dr. Kanarik and co-authors,

Thank you for the revised version of your manuscript. I am satisfied with your responses to the two reviews by the referees which I think were quite useful. Your manuscript is now accepted for publication in Ocean Science. Below I added a final list with technical comments. Please take this into account when submitting the final files of your manuscript.

Thank you for the acceptance of our manuscript and detailed corrections. We have provided answers to your points in below with blue colour.

 All through the manuscript the authors write magnitude(s) when the current velocity or current speed is meant. Please correct this.
In our manuscript we do not use the term magnitude to refer to any variable other than the current speed, and so we do not see a risk of misunderstanding here.  Hence, we do not see a reason to replace the term magnitude with the term current speed.  We only use the term velocity in a few places, where we intend to refer to both the magnitude and direction of the current.

The velocity should be shown with a space, like cm s-1. Please correct this throughout the manuscript. Corrected

L65 cm s-1 (insert space between cm and s) Corrected

 L66 more recent, instead of newer Corrected

L67 "they have been largely unpresented and unpublished before this paper" I suggest: they have not been public nor published before. Corrected, thank you for good suggestion

L70 highly complicated, instead of very complicated Corrected

L77 m s-1 (insert space between m and s) Corrected

L90 delete: In this paper, Corrected

L90 We have data, instead of We had data Corrected

Figure 1: Optionally, you may add the names of the different countries into the map. What are the grey lines near the top and bottom of the figure?

We choose not to include the names of the countries into the map, because it would require drawing also the borders. We think that the names of the different sea basins are enough to help readers to locate our study area.

The grey lines indicate the borders of different sea basins (named in the smaller map). This is now included in the figure caption.

Caption of Table 1: (last line): ... type of the region (SEE Section 3) where the measurementS were conducted. Corrected

L95 It is not clear what is meant with "ensembles". How is that different from just data? Please define ensembles. We changed this to data as using term ensemble served no important distinction in here.

L96 data were marked (data is plural) Corrected

L96 NaN (format) Corrected

L110 scatterers (typo) Corrected

L119 from 6 February to 22 March (format) Corrected

L154 "Because of this," instead of thus Corrected

L157 13 September 2006 and in Isokari AWS from 2 May 2006 (date format) Corrected

L165 publicly, instead of openly Corrected

L180-183 Is there any way that these regions can be shown in the figure? We coloured the names in different regions to Fig 1, to hopefully better show the groups of regions.

L187 Storms, instead of Wind storms Corrected

L192 I suggest: "3.1 Currents near the southern edge of the Archipelago Sea" or "3.1 Currents at the southern edge of the Archipelago Sea" Changed to "Currents near the southern edge of the Archipelago Sea"

L195 The mean current magnitudes (insert: current) Corrected

L196 The maximum current velocities ... (not: magnitudes) Corrected

L196 cm s-1 (space) Corrected

Figure 4: Caption, I suggest something like (if correct): Wind directions in different seasons in Isokari (a-d) and Utö winds (e-h) Changed to "Winds in different seasons in Isokari (a-d) and Utö winds (e-h)."

L200 in the layer closest to the surface, instead of: in the surfacmost layer Corrected

L275 northeastern AS (typo) Corrected

L281 "that cover all seasons", instead of: that range through all seasons Corrected

 L313 Do you mean? : During time periods when a seasonal thermocline was present … (same in line 316) Corrected

L524-528 This paragraph sounds very much like a conclusion and I suggest it should also appear there Thank you for noticing this. We moved these lines to the end of the conclusions and moved the notation in the beginning of this paragraph to the first paragraph of the conclusions.

 L530 delete: In this paper Deleted

 L571 Please translate the title of the publication and add it in brackets. Please also do that for some other references Corrected

With best wishes

Mario Hoppema